# On the Quality Assurance of Concept-based Representations

## Abstract

Recent work has focused on concept-based explanations, where deep learning models are explained in terms of high-level units of information, referred to as concepts. In parallel, the field of disentanglement learning has explored the related notion of finding underlying factors of variation in the data that have interpretability properties. Despite their overlapping purpose, the metrics to evaluate the quality of concepts and factors of variation in the two fields are not aligned, hindering a systematic comparison. In this paper, we consider factors of variation as concepts and thus unify the notations in concept and disentanglement learning. Next, we propose metrics for evaluating the quality of concept representations in both approaches, in the presence and in the absence of ground truth concept labels. Via our proposed metrics, we benchmark state-of-the-art methods from both families, and find that, contrary to common assumption, supervision alone may not be sufficient for quality assurance of concept representations. In light of this, we propose a set of guidelines to determine the impact that different degrees of supervision may have on the quality of learnt concept representations.

## 1 Introduction

Addressing the lack of interpretability of deep neural networks (DNNs) has given rise to explainability methods, most common of which are feature importance methods (Ribeiro et al., 2016; Lundberg & Lee, 2017) that quantify the contribution of input features to certain predictions (Bhatt et al., 2020). However, input features are not necessarily the most intuitive explanations, in particular when using low-level features such as pixels. *Concept-based explainability* (Ghorbani et al., 2019; Koh et al., 2020; Yeh et al., 2020; Ciravegna et al., 2021) remedies this issue by constructing an explanation at a concept level, where concepts are considered high-level and semantically meaningful units of information commonly used by humans to explain their decisions. Furthermore, concepts allow users to improve a model's performance via concept interventions, in which mispredicted concepts are corrected using expert knowledge (Koh et al., 2020).

In practice, what constitutes a concept is data-dependent, ranging from a group of pixels for image data (Ghorbani et al., 2019; Koh et al., 2020; Yeh et al., 2020), to a sequence of words and sub-graphs for text and graph-based data, respectively (Yeh et al., 2020; Magister et al., 2021). While all of these definitions are specific to the data modality, they commonly refer to an intermediate representation of the input data that has certain properties. Summarising concepts as intermediate representations of the data makes them analogous to factors of variation in *disentanglement learning*, where the assumption is that there exists a generative process capable of producing a high-dimensional dataset using a finite number of factors (Bengio et al., 2013). Such factors constitute a disentangled intermediate representation of the data with interpretability (Bengio et al., 2013; Higgins et al., 2017), fairness (Creager et al., 2019), and predictive performance (Locatello et al., 2019; 2020b) properties. The difference between Concept Learning (CL) and Disentanglement Learning (DGL) remains in that concepts in CL are often formed based on the supervision directly from concept labels or from a downstream task, whereas generative models (e.g., Variational Autoencoders (VAEs) (Kingma & Welling, 2014; Higgins et al., 2017)) that serve as the basis of DGL are un-/semi-supervised and factors of variation are directly informed by the distribution of the input data.

Since CL and DGL were developed independently, much of their connection remains unexplored. In particular, the metrics used to evaluate the quality of intermediate representations in each sub-

field are not aligned, despite their overlapping goals. Metrics in the concept literature (Koh et al., 2020; Kazhdan et al., 2020; Yeh et al., 2020) are mainly concerned with the properties of learnt concepts w.r.t. the downstream task. On the other hand, given the lack of a downstream task, metrics in disentanglement literature (Higgins et al., 2017; Ridgeway & Mozer, 2018; Locatello et al., 2019) are mainly concerned with the properties of the learnt representations, referred to as latent codes, w.r.t. the ground truth factors of variation. We argue that concepts/latent codes, as surrogate to inputs, need to have the following key properties: (i) They should correspond to semantically meaningful and coherent input sub-spaces; (ii) They should preserve the amount of mutual information observed in ground truth concepts or factors of variation (when available); and (iii) They should capture sufficient statistics to predict the downstream task (when available) as well as raw inputs do.

In this paper, we consider the properties (ii) and (iii) and make the following key contributions:

- We unify the language and notation across CL and DGL by framing factors of variation and latent codes in DGL as ground truth concepts and concept representations in CL, respectively.
- We introduce metrics for evaluating the quality of learnt concepts/codes in presence and absence of access to ground truth concepts/factors of variation and when concepts/codes are correlated.
- We conduct a systematic empirical comparison of state-of-the-art methods from four families of methods: supervised CL, unsupervised CL, semi-supervised DGL, and unsupervised DGL.
- We make the code used for our metrics, methods, and datasets available in an open-source library.[1]

## 2 BACKGROUND AND RELATED WORK

**Notation**  In both CL and DGL the aim is to find a low-dimensional intermediate representation $\hat{\mathbf{c}}$ that explains the downstream task(s) in CL, or the data's factors of variation in DGL. In CL, this low-dimensional representation corresponds to a matrix $\hat{\mathbf{c}} \in \hat{C} \subseteq \mathbb{R}^{d \times k}$ in which the $i$-th column constitutes a $d$-dimensional representation of the $i$-th concept. As zero-padding can be used to ensure equal length across different concept representations, for notational simplicity we assume that all concepts use a $d$-dimensional vector as their representation. Under this view, elements in $\hat{\mathbf{c}}_{(:,i)} \in \mathbb{R}^d$ are expected to have high values (under some reasonable aggregation function) if the $i$-th concept is considered to be activated for the input that generated this representation. For example, in the case where $d = T$ and each concept can take up to $T$ discrete values, $\hat{\mathbf{c}}_{(:,i)} \in [0,1]^T$ can represent a probability distribution over all values that the $i$-th concept can take. As most CL methods assume $d = 1$, for succinctness we use $\hat{c}_i$ in place of $\hat{\mathbf{c}}_{(:,i)}$ when $d = 1$.

We adopt the same representation for latent codes in DGL and let $\hat{\mathbf{z}} \in \hat{Z} \subseteq \mathbb{R}^{d \times k}$ be a latent code matrix such that each dimension $\hat{\mathbf{z}}_{(:,i)}$ (or a non-overlapping subset of dimensions) encodes one, and only one, independent factor of variation $z_j$. Nevertheless, note that in practice, $d$ tends to be 1 for most DGL methods. Finally, for simplicity, we use $\hat{\mathbf{c}}$ to refer to both learnt concept representations and latent codes. Ground truth concepts and factors of variations are referred to as $\mathbf{c} \in C \subseteq \mathbb{R}^k$.

In line with (Koh et al., 2020; Kazhdan et al., 2020; Yeh et al., 2020) we make use of: (i) a concept encoder function $g : \mathcal{X}' \mapsto \hat{C}$ that maps a transformation of the inputs $\mathbf{x} \in X \subseteq \mathbb{R}^m$, as performed by a function $\phi : X \mapsto \mathcal{X}'$, to a concept intermediate representation; and (ii) a label predictor function $f : \hat{C} \mapsto Y$ that maps the concept representations to a downstream task's set of labels $\mathbf{y} \in Y \subseteq \mathbb{R}^L$. These two functions can be combined to give a set of predictions for sample $\mathbf{x} \in X$ by computing $f(g(\phi(\mathbf{x})))$. In DGL autoencoders, one can think of $(g \circ \phi)(\cdot)$ as the autoencoder's encoder model and of the autoencoder's decoder model as a function that approximates $(g \circ \phi)^{-1}(\cdot)$.

**Supervised concept learning**  In supervised CL, access to concept labels $\mathbf{c}^{(i)} \in \mathbb{N}^k$, in addition to target labels $\mathbf{y}^{(i)} \in \mathbb{R}^L$, is assumed for inputs $\mathbf{x}^{(i)} \in \mathbb{R}^m$. In other words, we have training data $\{(\mathbf{x}^{(i)}, \mathbf{c}^{(i)}, \mathbf{y}^{(i)})\}_{i=1}^N$, where $N$ is the number of training samples. In its most common form, supervised CL divides the prediction into two distinct steps of: (i) mapping an input sample to its concept representation via a concept encoder $g$; and (ii) mapping a sample's concept representation to its task labels via a label predictor $f(\cdot)$. Together, these two functions constitute a *Concept Bottleneck Model* (CBM) (Koh et al., 2020), because their final prediction relies on the input going through the bottleneck $g(\phi(\mathbf{x}))$, which is trained to be component-wise aligned with $\mathbf{c}$.

---

[1]Code will be released after review.

*Concept-based Model Extraction* (CME) (Kazhdan et al., 2020) constructs a CBM from a pre-trained model by building a non-trivial $\phi(\cdot)$ mapping function using the model's latent space. Using such a latent representation instead of raw inputs typically makes CME more data efficient than CBM (Kazhdan et al., 2021). Similarly, *Concept Whitening* (CW) (Chen et al., 2020) constructs a CBM by introducing a pluggable batch normalization module whose activations $(g \circ \phi)(\cdot)$ are trained to be aligned with representative sets of binary concepts. It achieves this by forcing different feature maps of the normalization module to be decorrelated and orthogonal while incentivizing activations in a given axis to be high when its corresponding pre-defined concept is activated.

**Unsupervised Concept Learning** Unlike supervised CL, in unsupervised CL concept annotations are not available and concepts are discovered in an unsupervised manner. Ghorbani et al. (2019) extract concepts from a trained classifier for image data. Images belonging to each class are first segmented with multiple resolutions. The segments are then clustered as examples of class concepts and their importance scores are measured using TCAV (Kim et al., 2018).

Unlike Ghorbani et al. (2019), *Completeness-aware Concept Discovery* (CCD) (Yeh et al., 2020) is data modality agnostic and extracts class-independent concepts. CCD builds on TCAV to first extract a set of concept vectors $\{\mathbf{h}^{(i)}\}_{i=1}^{k}$, each of which is a unit vector in $\mathcal{X}'$. These vectors are then used to construct a concept representation $g(\phi(\mathbf{x})) = \hat{\mathbf{c}} \in \hat{C} \subseteq \mathbb{R}^k$ by setting $\hat{c}_i$ to $TH(\langle \phi(\mathbf{x}), \mathbf{h}^{(i)} \rangle, \beta)$, the $\beta$-thresholded inner product of $\phi(\mathbf{x})$ and concept vector $\mathbf{h}^{(i)}$. When *complete*, a concept representation should contain sufficient statistics to obtain a high performance in the original model's prediction task $f(\phi(\mathbf{x}))$. If this is the case, then there must exist a mapping $\psi : \hat{C} \mapsto \mathcal{X}'$ that recovers $\phi(\mathbf{x})$ from $g(\phi(\mathbf{x}))$. Thus, CCD constructs a set of concept vectors such that $f(\psi(g(\phi(\mathbf{x}))))$ is able to achieve a similar task performance to that of $f(\phi(\mathbf{x}))$. Notice that although $f(\cdot)$ is still a label predictor function here, it is not applied to the raw inputs or to their concept representation.

Similarly, *Self-Explainable Neural Networks* (SENNs) (Alvarez-Melis & Jaakkola, 2018) learn to produce concept-based explanations without explicit concept supervision through a robustness regularization term that encourages a differentiable model to locally act linearly. It proceeds by first learning a concept representation $g(\mathbf{x}) = \hat{\mathbf{c}} \in \mathbb{R}^k$ from the encoder of an autoencoding model and, second, generating a prediction $f(\hat{\mathbf{c}}) = \mathcal{G}(\theta(\mathbf{x})_1 \hat{c}_1, \cdots, \theta(\mathbf{x})_k \hat{c}_k)$ using an aggregation function $\mathcal{G}$ to weight the importance of each concept with a linear coefficient learnt through a differentiable model $\theta(\cdot)$. Concept weights $\theta(\mathbf{x})$ can then serve as an explanation for SENN's predicted label.

**Disentanglement Learning** Generative models (e.g., VAEs (Kingma & Welling, 2014)) used in DGL assume that data is generated from a set of independent factors of variation $\mathbf{c} \in C$, sampled from factorable distribution $p(\mathbf{c}) = \prod_i p(c_i)$, such that a sample $\mathbf{x}$ is generated according to the conditional distribution $p(\mathbf{x}|\mathbf{c})$. Thus, the goal of DGL is to find a function $g(\cdot)$ that maps inputs to a disentangled latent representation, such that a subset of non-overlapping dimensions of the latent representation corresponds to a unique factor of variation $c_i$.

In light of recent work showing the theoretical impossibility of learning disentangled representations in an unsupervised manner (Locatello et al., 2019), a promising line of work suggests providing the inductive bias required to learn disentangled representations through weak supervision. In this work, we focus on DGL methods where weak supervision comes via pairs of observations whose corresponding ground truth factors of variation share at least one common element (Locatello et al., 2020a) and contrast them against vanilla unsupervised DGL methods such as VAEs (Kingma & Welling, 2014) and $\beta$-VAEs (Higgins et al., 2017).

## 3 QUALITY ASSURANCE OF CONCEPTS

Given the varying degree of supervision in the approaches above, the quality of their concept representations is expected to vary. The quality of learnt concepts w.r.t. downstream tasks has commonly been studied using predictive performance metrics such as accuracy and AUC. Similarly, there exist several metrics for measuring the properties of learnt concepts w.r.t. the ground truth ones, either focusing on the *overall set of learnt concepts* or focusing on *individual concepts* (see Appendix 6.1 for a summary). Nevertheless, these metrics have two main shortcomings.

The first shortcoming is assuming that concepts are independent, implying that if a concept representation encodes information beyond that of the ground truth concept it is aligned with, it counts as

*leakage* (Mahinpei et al., 2021). Impurities caused by leakage can have far-reaching consequences in terms of human interventions, e.g., perturbations of an impure concept may impact other concepts unintentionally. While we agree with the danger of impurities, we argue that concepts in the real-world often have dependencies, and full disentanglement is not realistic. As such, leakage as described above is not necessarily undesirable when it reflects the correlations found in ground-truth concepts. To support this, we demonstrate the lack of inter-concept independence in a real-world dataset, the Caltech-UCSD Birds-200-2011 (CUB) (Wah et al., 2011a), in Appendix 6.2.

The second shortcoming is the assumption that one has access to ground truth concepts, which are used to verify the properties of the extracted ones. Previous work has shown that access to ground truth concepts is limited given the difficulty of annotating samples with concept labels (Raghavan et al., 2006). The rarity of real-world datasets with such annotations is evidence for this claim.

To address these shortcomings, we first propose the *oracle impurity score*, a metric that measures the quality of concept representations when one has access to discrete ground truth concepts labels which may be correlated. We then propose the *niching purity score* and *niching impurity score*, two efficient metrics which, in conjunction, measure the quality of concept representations when one has only access to a classification task's labels.

### 3.1 ORACLE IMPURITY

To circumvent the need for inter-concept independence, we take inspiration from (Mahinpei et al., 2021), where they informally measure concept impurity as how predictive a CBM-generated concept probability is for the ground truth value of other independent concepts. If the pre-defined concepts are independent, then the inter-concept predictive performance should be no better than random. To generalise this assumption beyond independent concepts, we first measure the predictability of ground truth concepts w.r.t. one another. Then we measure the predictability of learnt concepts w.r.t. the ground truth ones. The divergence between the former and the latter acts as an impurity metric, measuring the amount of undesired information that is encoded by the learnt concepts. In order to formally introduce our metric, we begin by formalising a *purity matrix*.

**Definition 1** (Purity Matrix)**.** *Given concept representations* $\hat{\Gamma} = \{\hat{\mathbf{c}}^{(i)} \in \mathbb{R}^{d \times k}\}_{i=1}^{n}$*, and corresponding discrete ground truth concept annotations* $\Gamma = \{\mathbf{c}^{(i)} \in \mathbb{N}^{k}\}_{i=1}^{n}$*, assume that* $\hat{\Gamma}$ *and* $\Gamma$ *are aligned element-wise: for all* $l \in \{1, \cdots, k\}$*, the* $l$*-th concept representation of* $\hat{\mathbf{c}}^{(i)}$ *encodes for the same concept as the* $l$*-th concept label in* $\mathbf{c}^{(i)}$*. The Purity Matrix of* $\hat{\Gamma}$ *given ground truth labels* $\Gamma$ *is defined as a matrix* $\pi(\hat{\Gamma}, \Gamma) \in [0, 1]^{k \times k}$ *whose entries are given by:*

$$\pi(\hat{\Gamma}, \Gamma)_{(i,j)} := AUC\big(\big\{\big(\psi_j(\hat{\mathbf{c}}_{(:,i)}^{(1)}), c_j^{(1)}\big), \cdots, \big(\psi_j(\hat{\mathbf{c}}_{(:,i)}^{(n)}), c_j^{(n)}\big)\big\}\big)$$

*where* $\psi_j(\cdot)$ *is a classifier (e.g., an MLP) trained to map the* $i$*-th concept's* $d$*-dimensional vector representation* $\hat{\mathbf{c}}_{(:,i)}$ *to a probability distribution over all the values that concept* $j$ *may take.*

The $(i, j)$-th entry of $\pi(\hat{\Gamma}, \Gamma)$ contains the AUC when predicting the ground truth value of concept $j$ given the $i$-th concept representation. The diagonal entries of this matrix show how good a concept representation is at predicting its corresponding ground truth value, while the off-diagonal entries show how good such a representation is at predicting the ground truth labels of other concepts. Intuitively, one can think of the $(i, j)$-th entry of this matrix as a proxy of the mutual information between the $i$-th concept representation and the $j$-th ground truth concept. While in principle several other methods could have been used to estimate this mutual information, we choose AUC primarily for two reasons: (1) it is computationally tractable and (2) it allows the construction of a metric that can be easily bounded (as discussed below). Furthermore, while in this work, as in the majority of the existing CL literature (Koh et al., 2020; Chen et al., 2020; Yeh et al., 2020), we focus on concepts that are binary in nature, notice that one can trivially extend our definition to be applicable to multivariate concepts by using the mean one-vs-all AUC across all possible labels a ground truth concept may take. Details on how this matrix is computed in practice can be found in Appendix 6.3. This definition of a purity matrix allows us to construct a metric for quantifying the amount of unnecessary information, or impurity, that a concept encoder encodes.

**Definition 2** (Oracle Impurity Score (OIS))**.** *Let* $g : X \mapsto \hat{C} \subseteq \mathbb{R}^{d \times k}$ *be a concept encoder and let* $\Gamma_X := \{\mathbf{x}^{(i)} \in X\}_{i=0}^{n}$ *and* $\Gamma := \{\mathbf{c}^{(i)} \in \mathbb{N}^{k}\}_{i=0}^{n}$ *be ordered sets of testing samples and*

*their corresponding concept annotations, respectively. If, for any ordered set $A$ we define $g(A)$ as $g(A) := \{g(a) \mid a \in A\}$, then the OIS of $g(\cdot)$ given data $(\Gamma_X, \Gamma)$ is defined as:*

$$OIS(g, \Gamma_X, \Gamma) := \frac{2\left\|\pi\big(g(\Gamma_X), \Gamma\big) - \pi\big(\Gamma, \Gamma\big)\right\|_F}{k}$$

*where we use $||\mathbf{A}||_F$ to represent the Frobenius norm of matrix $\mathbf{A}$.*

Intuitively, the OIS measures the total deviation of an encoder's purity matrix with the purity matrix obtained from using the ground truth concept labels only (i.e., the "oracle matrix"). We opt to measure this divergence using the Frobenius norm of their difference, instead of other similarity metrics, in order to obtain a bounded output which can be easily interpreted. More specifically, since each entry in the difference $\big(\pi\big(g(\Gamma_X), \Gamma\big) - \pi\big(\Gamma, \Gamma\big)\big)$ can be at most $1/2$, the upper bound of the Frobenius norm of this difference is $k/2$. Therefore, our metric includes a normalisation factor $2/k$ which guarantees that it is in $[0, 1]$. This allows interpreting an OIS of $1$ as a complete misalignment between $\pi\big(\Gamma, \Gamma\big)$ and $\pi\big(g(\Gamma_X), \Gamma\big)$ (i.e., the $i$-th concept representation can predict all other concept labels except its corresponding one even when concepts are independent). An impurity score of $0$, on the other hand, represents perfect alignment between the two purity matrices (i.e., the $i$-th concept representation does not encode any unnecessary information for predicting concept $i$).

### 3.2 NICHING-BASED METRICS

In nature, a niche is defined as a resource-constrained subspace of the environment that can support different types of life (Darwin, 1859). By analogy, in the context of neural networks each layer can be seen as a computational "environment" with a finite amount of computational resources (i.e., the layer parameters). A "niche of neurons" can be defined as a subspace of the layer (i.e., the "environment") that is critical for the accurate prediction (i.e., "the survival") of a specific downstream task (i.e., the "species living in the niche").

Inspired by the theory of niching, concept bottleneck sensitivity, and sparsity (Dimanov & Jamnik, 2018; Barbiero et al., 2021), we propose two metrics that measure the quality of concept representations in the *absence* of access to ground truth concept labels and in the presence of task-specific labels. Our metrics quantify how task-label-separable a concept representation is. For this, they first identify a set of concepts that are important for a task label, and then see if such a set, referred to as the label's *concept niche*, is complete enough to predict the label. The predictive performance of a set of concepts outside the concept niche of a label identifies impurity and unnecessary leakage in the concept representation. The existence of niches is not an assumption for the metrics to be viable in practise, but rather an observation in many real-world data (see Appendix 6.6 for an example). We start by describing a *concept nicher*, a function that defines the dependency between a single concept and a single task label.

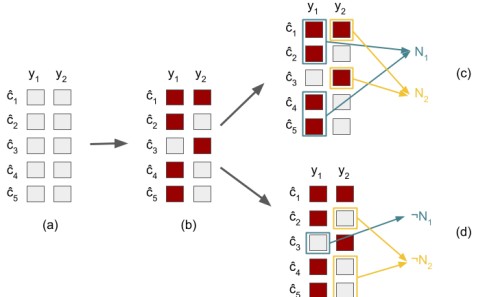

Figure 1: Concept niches and their complement: (a) given a set of concepts and a set of classification labels, (b) a concept nicher identifies the concepts on which a label depends strongly (red blocks); thus (c) a concept niche is the set of these concepts that a label depends on strongly; and (d) a complement concept niche is the set of concepts that a label does not depend on strongly.

**Definition 3** (Concept nicher). *Given a label predictor $f : \hat{C} \mapsto Y$, where $\hat{C} \subseteq \mathbb{R}^{d \times k}$ and $Y \subseteq \mathbb{R}^L$, we define a concept nicher as a Boolean-valued function $\nu_f : \{1, \cdots, k\} \times \{1, \cdots, L\} \mapsto \{0, 1\}$ that returns $\nu_f(i, j) = 1$ if the $j$-th output dimension of $f(\cdot)$ has a "strong" dependency, as determined by a reasonable procedure, on the $i$-th concept representation $\hat{\mathbf{c}}_{(:,i)}$ and $0$ otherwise.*

The above definition is affected by how a dependency is defined. One efficient way of measuring the dependency, and the way that we proceed, is to use the absolute Pearson correlation, denoted as $\rho$. We call such an instantiation a *concept-correlation nicher* (CCorrN) and define it as:

$$\text{CCorrN}_f(i, j) := \left|\rho\big(\{\hat{\mathbf{c}}_{(:,i)}^{(1)}, \cdots, \hat{\mathbf{c}}_{(:,i)}^{(N)}\}, \{f(\hat{\mathbf{c}}^{(1)})_j, \cdots, f(\hat{\mathbf{c}}^{(N)})_j\}\big)\right| > \beta$$

where $\beta \in [0,1]$ is a user-defined threshold. If $\hat{\mathbf{c}}_{(:,i)}$ is not a scalar representation (i.e., $d > 1$), we use the maximum absolute correlation coefficient between all entries in $\hat{\mathbf{c}}_{(:,i)}$ and the target label $f(\mathbf{c})_j$ as a representative correlation coefficient for the entire representation $\hat{\mathbf{c}}_{(:,i)}$. For all experiments we use $\beta = 0.2$. A study of the impact of $\beta$ on CCorrN is in Appendix 6.5.

For each task label, the *concept niche* defines the set of concepts on which the label depends (i.e., where the concept nicher outputs 1 for the label). One can intuitively see this in Figure 1(b), where the red blocks show the concepts that the concept nicher has identified for $y_1$ and $y_2$ and Figure 1(c) shows the niches. Formally we define a concept niche as:

**Definition 4** (Concept niche). *The concept niche $N_j(\nu_f)$ for label $j$, determined by label predictor $f(\cdot)$ and concept nicher $\nu_f(\cdot)$, is defined as $N_j(\nu_f) := \big\{ i \mid i \in \{1, \cdots, k\} \text{ and } \nu_f(i,j) = 1 \big\}$.*

Abusing notation, we let $\neg N_j(\nu_f) := \{1, \cdots, k\} \setminus N_j(\nu_f)$ be the complement of the set $N_j(\nu_f)$ (see Figure 1(d)). Although the label predictor $f$ can be any classifier, in our experiments we use a ReLU MLP with hidden layer sizes $\{20, 20\}$. In this setup, we let $f|_{N_j(\nu_f)}$ be the MLP resulting from pruning all connections in $f$ except those between concept representations in the niche $N_j(\nu_f)$ and the $j$-th output of $f$.

We define the *Niche Purity Score* (NPS) and *Niche Impurity Score* (NIS) as measures of the predictive capacity of a niche and its complement w.r.t. a specific classification label for a given label predictor function, respectively. A good concept representation has a simultaneous high NPS and low NIS for each label, as high values in the latter score are a symptom of unnecessary leakage in concept representation.

**Definition 5** (Niche Purity Score). *Given label predictor $f : \hat{C} \mapsto Y$, concept nicher $\nu_f$, concept representations $\hat{\Gamma} = \{\hat{\mathbf{c}}^{(j)}\}_{j=1}^n$, and corresponding ground truth classification labels $\Gamma_Y = \{\mathbf{y}^{(j)}\}_{j=1}^n$, the NPS of the $i$-th output of $f(\cdot)$ is defined as $NPS_i(f, \nu_f) := AUC\big(\{(f|_{N_i(\nu_f)}(\hat{\mathbf{c}}^{(j)}_{(:,N_i(\nu_f))}), y_i^{(j)})\}_{j=1}^n\big)$. The NPS of $f$, defined as $NPS(f, \nu_f) := \sum_{i=1}^L NPS_i(f, \nu_f)/L$, is the mean NPS across all labels.*

**Definition 6** (Niche Impurity Score). *The NIS of concept niche $N_j(\nu_f)$ is the NPS of $\neg N_j(\nu_f)$.*

Essentially, the NPS metric measures the quality of niche $N_i(\nu_f)$ via the AUC of predicting the ground truth label of the $i$-th label using *only* the concept representations corresponding to niche $N_i(\nu_f)$. A NPS score of 1 (its maximum value), thus indicates a perfect niche purity for the $i$-th output of function $f(\cdot)$ while a worst-case-scenario NPS of $1/2$ indicates that a niche is unable to do better than random when predicting its corresponding label. Similarly, the NIS metric measures the impurity of niche $N_i(\nu_f)$ by measuring the AUC of predicting the ground truth label of the $i$-th label using concept representations *outside* the niche corresponding to the $i$-th label. Therefore, a NIS score of $1/2$ indicates the best-case scenario impurity as concepts outside of niche $N_i(\nu_f)$ hold no useful information to predict the $i$-th label. In contrast, a worst-case NIS score 1 corresponds to complete impurity as the $i$-th label can be perfectly predicted from concepts that are irrelevant to it.

## 4 EXPERIMENTS

**Methods, Model Selection, and Training** We compare the quality of concept representations in various methods using our metrics. CBM (Koh et al., 2020) is selected from the supervised CL family due to its fundamental role in CL. We focus on jointly trained CBMs, where the task-specific loss and the concept prediction loss are minimised jointly. We also add CW (Chen et al., 2020) from this family as it directly treats decorrelated concept representations. From the unsupervised CL family, CCD (Yeh et al., 2020) is selected due to its data agnostic nature and SENN (Alvarez-Melis & Jaakkola, 2018) due to its particular mix of ideas from both DGL and CL literature. From the DGL family, we consider two weakly supervised methods, Adaptive Group Variational Autoencoder (Ada-GVAE) and Adaptive Multilevel Variational Autoencoder (Ada-MLVAE) (Locatello et al., 2020a), as well as two unsupervised methods, namely vanilla Variational AutoEncoders (VAE) (Kingma & Welling, 2014) and $\beta$-VAE (with $\beta = 10$) (Higgins et al., 2017). For each method and metric, we report the average metric values and $95\%$ confidence intervals obtained from 5 different random seeds. Given that the number of concepts in unsupervised CL and DGL approaches is not known upfront, we allow an extra variation, where the number of learnt concepts is twice the number

of ground truth ones. We observe that model capacity, in particular the encoder's capacity, impacts the quality of learnt concepts (see Appendix 6.7). Therefore, to minimise the effects of inductive biases arising from the model architecture and training hyperparameters, we use the same architecture and training setups for all methods within the same dataset whenever possible. We include details on training and architecture hyperparameters in Appendix 6.8.2.

**Datasets** In order to have datasets that are compatible with both CL and DGL, we construct datasets whose samples are fully described by a vector of ground truth generative factors. Moreover, to simulate real-world scenarios, we work with tasks with a varying degree of dependencies in their concept annotations. To achieve this, we first design a parametric binary-class dataset *TabularToy*($\delta$), a variation of the tabular dataset proposed by (Mahinpei et al., 2021). We also construct two multiclass image-based parametric datasets: *dSprites*($\lambda$) and *3dshapes*($\lambda$), based on dSprites (Matthey et al., 2017) and 3dshapes (Burgess & Kim, 2018) datasets, respectively. They consist of 3D samples generated from a vector consisting of $k = 5$ and $k = 6$ ground truth factors of variation, respectively. Both datasets have one binary concept annotation per factor of variation. Parameters $\delta \in [0, 1]$ and $\lambda \in \{0, \cdots, k-1\}$ control the degree of concept inter-dependencies during generation (a value of 0 represents inter-concept independence while higher values represent stronger inter-concept dependencies). Details on how dependencies are introduced, as well as the nature of the individual concepts and task labels for all our datasets, can be found in the Appendix 6.8.1.

## 4.1 RESULTS

**Overall set of learnt concepts is equally predictive of the downstream task across all surveyed methods.** The quality of the overall set of learnt concepts w.r.t. downstream tasks, measured by its mean one-vs-all task AUC, does not vary considerably across methods and stays close to $100\%$. This result replicates the predictive power of the raw inputs and thus shows that the grouped concept representations faithfully capture task-related information in the input space. Plots for predictive AUC of raw inputs and the overall set of learnt concepts can be found in Appendix 6.9.

**Supervision may not lead to overall set of learnt concepts being more predictive of individual ground truth concepts compared with no supervision (Figure 2).** Since CBM and CW benefit from explicit concept supervision, we expect their learnt representations to predict the ground truth concepts well. While this observation holds in CBM and CW when one uses CW's entire feature maps as concept representations (shown as "CW Feature Map" in plots), it does not hold for "CW MaxPool-Mean," where one reduces the CW module's feature maps to scalar concept scores using a max pool followed by a mean reduction (as seen in Yeh et al. (2020)). This is in alignment with results reported in Mahinpei et al. (2021), and suggests that there may be substantial information loss when reducing CW's feature maps into scalar representations. Note that this is not observed in ToyTabular($\delta$) as its concept representations are already scalars. Interestingly, the implicit supervision (i.e., supervision from downstream task as opposed to explicit concept supervision) in unsupervised CL can be just as effective as explicit concept supervision in other methods in terms of the predictive power of the overall set of learnt concepts for individual ground truth ones. This is observed in CCD and to a lesser degree in SENN. We attribute the gap between the two to CCD being designed to maximize the completeness scores of its overall set of learnt concepts w.r.t. the task. Surprisingly, complete lack of supervision in DGL's VAE and $\beta$-VAE appears to be more effective than the weak supervision in Ada-MLVAE and Ada-GVAE. This is unexpected given that such weak supervision was introduced to address the difficulty of learning representations that capture the data's underlying factors of variations in an unsupervised manner (Locatello et al., 2019).

**Metrics that assume concept independence may be misleading (Figure 3).** The comparisons above focused on the quality of overall set of learnt concepts, without an indication of the quality of individual learnt concepts w.r.t. the ground truth ones. The alignment between the learnt and ground truth concepts is clear in CBM and CW. In CCD and DGL, where there is no alignment, we compute a greedy alignment between the learnt and ground truth concepts based on the predictive AUC of using a learnt concept to predict each ground truth concept (i.e., a ground truth concept is assumed to be represented by the learnt one that can predict it best). Details on this are in Appendix 6.4.

After the alignment, we investigate the quality of individual learnt concepts w.r.t. the ground truth ones. First, we show the inadequacy of assuming concept independence compared to our oracle impurity metric proposed in Definition 2. We achieve this by comparing our metric against a variant

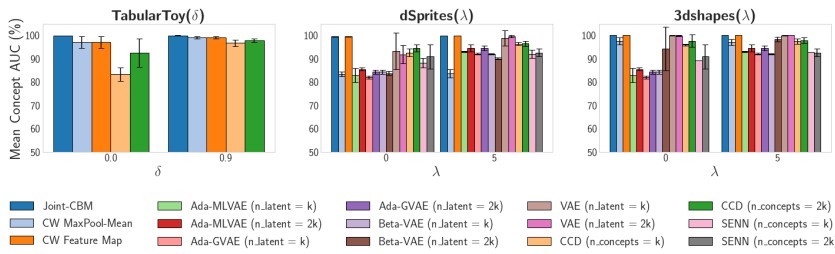

Figure 2: Mean concept AUC calculated by averaging over all the AUCs of predicting individual ground-truth concepts from the entire set of learnt concepts. Each AUC is calculated using a ReLU MLP with hidden layers $\{64, 64\}$ that is trained to predict the target concept.

of itself, labeled as "non-oracle impurity", that assumes the oracle matrix of any dataset is always a matrix with $1$ in its diagonals and $1/2$ in its off-diagonals. We observe that the non-oracle impurity misleadingly shows more impurity as the degree of inter-concept dependencies increases. This is due to the fact that our metric, in contrast to the non-oracle impurity, takes into account the dataset's inherent ground truth inter-concept dependencies as a baseline for what a method's purity matrix should look like. We observe the same results across other methods (see Appendix 6.10).

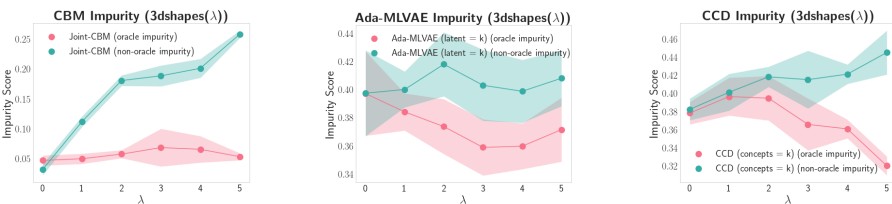

Figure 3: Impurity scores (both oracle and non-oracle) in *3dshapes*$(\lambda)$.

**Oracle impurity demonstrates that explicit or implicit supervision may not result in purer individual concepts than no supervision (Figure 4).** The oracle impurity metric results demonstrate that CBM's individual concepts consistently experience the least amount of impurity due to receiving explicit supervision, which is to be expected. Unexpectedly, though, we observe that the same explicit concept supervision can lead to highly impure representations: CW's full feature maps concepts encode the highest amount of impurity on average, a phenomenon observed by Mahinpei et al. (2021) too. This is due to the fact that the learnt concepts are highly predictive of every ground truth one irrespective of their alignment. Thus, while explicit supervision translates to better concept predictive performance, it does not necessarily translate to purer concepts. Looking into implicit supervision, while the overall set of learnt concepts in CCD performs as well as those learnt via explicit concept supervision in CBM and CW, individual concepts do not correspond well to the ground truth ones. This indicates that the information about each ground truth concept is distributed across the overall representation rather than localized to individual concepts. As a result, surprisingly, in absence of extra capacity, CCD and SENN exhibit an OIS in a similar or better range than methods benefiting from explicit supervision such as CW. We attribute CCD's lower impurity, compared with SENN, to the use of a regularization term that encourages coherence between concept representations in similar samples and misalignment between concept representations in dissimilar samples. More interestingly, however, SENN encodes higher impurities in its concept representations than all DGL approaches despite benefiting from explicit supervision. Within DGL approaches, not only does the lack of supervision result in better predictive performance for the overall set of learnt concepts as discussed before, but it astonishingly also results in lower impurity representations than those of weakly-supervised DGL methods.

**Niching-based metrics demonstrate that supervision may not result in task-label-separable concept representations compared to no supervision (Figure 4).** Niching-based metrics do not differentiate between explicit and implicit supervision as much as oracle impurity, as they merely focus on task labels that are available in both approaches. Thus, simultaneous low niche impurity and high niche purity is observed in CBM and CCD, despite their differing degree of supervision. There are, however, surprising exceptions to this: in contrast to CBM and CCD, high impurity and

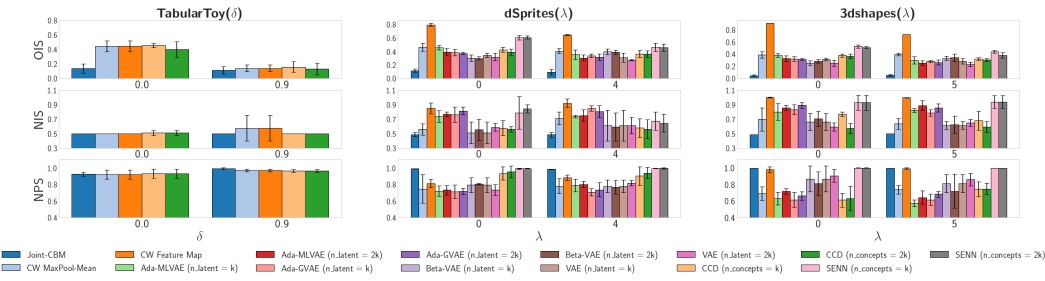

Figure 4: Evaluation of different models, deployed across various tasks, using our proposed metrics.

high purity are observed in CW's full feature maps (i.e., when no scalar reduction is applied), as well as in SENN's representations. This is particularly true when concept dependencies increase, which shows the vulnerability of this approach when concepts have inter-dependencies. Performing a MaxPool-Mean reduction in CW's maps reduces the amount of impurities, however, it also reduces the amount of purity (i.e., the learnt concepts are unable to predict their aligned ones with high confidence). Similar to oracle impurity, this is further evidence that some crucial information may be lost in CW's scalar reduction, confirming our OIS result. Within DGL approaches, not only does the lack of supervision results in better predictive performance for the overall set of learnt concepts, but it surprisingly also results in lower impurity and higher purity that those of weakly-supervised DGL, a phenomenon worth investigating in the future.

**Oracle impurity and niching-based metrics are robust to concept inter-dependencies.** The highly preserved ranking of the methods using our metrics in both settings with and without dependencies, indicates the robustness of these metrics to concept inter-dependencies as intended.

## 5 DISCUSSION AND CONCLUSION

The lack of metrics to evaluate and contrast the quality of concept representations within and between CL and DGL methods hinders their systematic comparison. In this paper, we address this limitation by introducing a set of metrics that facilitate comparison from concept representation quality perspective. Our systematic evaluation of several methods using our metrics suggests that: (i) Model design is just as important as degree of supervision: models with the same degree of supervision (e.g., CBM vs. CW and CCD vs. SENN) can encode varying amounts of impurity in their concept representations, in particular when there are strong cross-concept dependencies; (ii) In settings where ensuring the quality of the overall set of learnt concepts is sufficient, implicit supervision from task labels can be just as effective as explicit supervision from concept labels. Thus, given the cost, providing explicit supervision is not recommended; (iii) In settings where ensuring the quality of individual concept representation as well as their overall set is required, explicit supervision as used in CBM is the most recommended option, followed by CCD if the degree of cross-concept dependencies in the data is unknown; and (iv) In either of the above settings the weak supervision provided by DGL methods is not recommended as a substitute for implicit or explicit supervision. In fact, overall DGL methods without any supervisions tend to outperform the weakly supervised ones. Related to metrics, in setting (ii), regardless of access to concept labels, niching-based metrics should be favoured to oracle impurity. Both metrics are able to capture that the overall set of learnt concept benefits from supervision, with the former metric being very efficient and tractable when the number of concepts is high (as it does not require a classifier to be trained for every pair of concepts). On the contrary, in setting (iii), the high computational cost of oracle impurity is justified if concept labels are available. This is because, unlike niching-based metrics, oracle impurity distinguishes the impact of explicit vs. implicit supervision on the quality of individual learnt concepts.

One direction that was not explored in this paper is the evaluation of the quality of concept representations w.r.t. the input space. In fact, Margeloiu et al. (2021) showed that concepts from CBM often do not correspond to semantically meaningful input space. We therefore believe that future work on defining metrics that capture concept-input quality along with the metrics we proposed here can provide a more comprehensive assurance about the quality of learnt concept representations.

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

# 6 Additional Resources

## 6.1 Metrics Related to Properties of Concept Representations w.r.t. Ground Truth Concepts

Refer to Table 1 for a summary of some metrics used for measuring the quality of disentangled representations which are applicable to disentangled concept representations.

Table 1: Metrics related to properties of concepts w.r.t. ground truth concepts divided to two categories: those that capture properties of individual concepts and those that capture properties of the overall set of learnt concepts w.r.t. ground truth concepts. Although metrics in each column serve the same purpose, they are mathematically distinct.

| Individual learnt concepts w.r.t. ground truth concepts | Overall set of learnt concepts w.r.t. ground truth concepts |
|---|---|
| Modularity (Ridgeway & Mozer, 2018): Whether each learnt concept corresponds to at most one ground truth one. Measured by deviation from ideal case where each learnt concept has high mutual information with only one ground truth one and zero with others. | Explicitness (Ridgeway & Mozer, 2018): Whether the overall set of learnt concepts can predict each individual ground truth ones using a simple (e.g., linear) classifier. Measured by the average predictive performance of concept vector. |
| Mutual information gap (Chen et al., 2018): Whether learnt concepts are disentangled. Measured by averaging the gap in mutual information between the two learnt concepts that have the highest mutual information with a ground truth concept. This metric generalises the disentanglement scores in Higgins et al. (2017) and Kim & Mnih (2018). | Informativeness (Eastwood & Williams, 2018): Whether the overall concept vector can predict each ground truth concept with low prediction error. Measured by average prediction error of concept vector. |
| Disentanglement (Eastwood & Williams, 2018): Whether each learnt concept captures at most one ground truth one. Measured by the weighted average of disentanglement degree of each learnt concept. Such degree is calculated based on entropy, where high entropy for a learnt concept shows its equal importance for all ground truth ones and therefore its low disentanglement degree. The weight is calculated based on the aggregation of relative importance of a learnt concept in predicting each ground truth one. | |
| Alignment (Yeh et al., 2020): Whether the learnt concepts match the ground truth ones. Measured by average accuracy of predicting each ground truth concept using the learnt one that predicts it best. | |

## 6.2 Mutual Information in CUB Dataset Concepts

We take the Caltech-UCSD Birds-200-2011 (CUB) dataset (Wah et al., 2011a), formed by images of 200 types of birds, and use each sample's 112 binary attributes as concept annotations (Koh et al., 2020). We find the top-10 concepts with highest mutual information with the label and compute their Pearson correlation coefficients. Our results, shown in Figure 5, highlight that many concepts have non-zero correlations. This shows the lack of inter-concept independence in real-world datasets.

## 6.3 Purity Matrix Implementation Details

We compute the (i, j)-th entry of the purity matrix as follows: we split the original testing data $(X_{\text{test}}, Y_{\text{test}}, C_{\text{test}})$ into two disjoint sets, a new training set $(X'_{\text{train}}, Y'_{\text{train}}, C'_{\text{train}})$ and a new testing set $(X'_{\text{test}}, Y'_{\text{test}}, C'_{\text{test}})$, using a traditional 80%-20% split. We then use the concept representations learnt

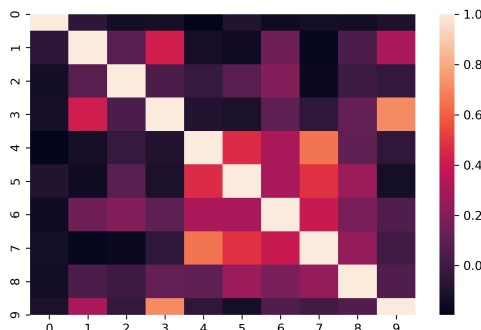

Figure 5: Absolute Pearson correlation coefficients of the top-10 concepts with highest label mutual information in CUB dataset. Notice that there exist strong correlations between some of the concepts.

for the $i$-th concept for samples in $X'_{\text{test}}$ to train a ReLU MLP $\psi_j(\cdot)$ with a single hidden layer with 32 activations to predict to truth value of the $j$-th ground-truth concept. In other words, we train $\psi_j(\cdot)$ using labelled samples $\left\{ \left( g\big(\phi(\mathbf{x}^{(l)})\big)_{(:,i)}, \mathbf{c}_j^{(l)} \right) \mid \mathbf{x}^{(l)} \in X'_{\text{train}} \wedge \mathbf{c}^{(l)} \in C'_{\text{train}} \right\}$. Finally, we set the $(i,j)$-th entry of the purity matrix as the AUC achieved when evaluating $\psi_j(\cdot)$ on the new testing set $\left( g\big(\phi(X'_{\text{test}})\big), C'_{\text{test}} \right)$.

## 6.4 Concept Alignment in Unsupervised Concept Representations

In the presence of ground truth concept annotations, one can still compute a purity matrix, and therefore the OIS, even when the concept representations being evaluated were learnt without direct concept supervision. We achieve this by finding an injective alignment $\mathcal{A} : \{1, 2, \cdots, k\} \mapsto \{1, 2, \cdots, k'\}$ between ground truth concepts $\mathbf{c} \in \mathbb{R}^k$ and learnt concept representations $\hat{\mathbf{c}} \in \mathbb{R}^{d \times k'}$. In this setting, we let $\mathcal{A}(i) = j$ represent the fact that the $i$-th ground truth concept $c_i$ is best represented by the $j$-th learnt concept representation $\mathbf{c}_{(:,j)}$. In our work, we greedily compute this alignment starting from a set of unmatched ground truth concepts $\mathcal{I}_{\text{ground}}^{(0)} = \{1, \cdots, k\}$ and a set of unmatched learnt concept representations $\mathcal{I}_{\text{learnt}}^{(0)} = \{1, \cdots, k'\}$ and iteratively constructing $\mathcal{A}$ by adding one match $(i, j)$ at a time. Specifically, at time-step $t+1$ we match ground truth concept $i$ with learnt concept $j$ if one can predict concept $i$ from $\mathbf{c}_{(:,j)}$ better than every other concept representation $\mathbf{c}_{(:,j')}$ can predict every other ground truth concept $c_{i'}$. We evaluate predictability of ground truth concept $c_i$ from learnt concept representation $\hat{\mathbf{c}}_{(:,j)}$ by training a ReLU MLP with a single hidden layer with 32 activations and evaluating its AUC on a test set. Once a match between ground truth concept $i$ and learnt concept representation $j$ has been established, we set $\mathcal{I}_{\text{ground}}^{(t+1)} := \mathcal{I}_{\text{ground}}^{(t)} \setminus \{i\}$ and $\mathcal{I}_{\text{learnt}}^{(t+1)} := \mathcal{I}_{\text{learnt}}^{(t)} \setminus \{j\}$. We repeat this process until we have found a match for every ground truth concept (i.e., until $\mathcal{I}_{\text{ground}}^{(t)}$ becomes the empty set). Notice that in practice, one needs to compute the predictability of ground truth concept $i$ from concept representation $j$ only once when building alignment $\mathcal{A}$.

## 6.5 Impact of CCorrN's $\beta$ on Niche Purity, Impurity, and Size

$\beta$ is not a parameter to tune, it is rather a knob to be used to assess the stability and robustness of the concept representations. To this aim, we compute niching scores for different values of $\beta$ and generate a plot as reported in Figure 6. The figure shows the impact of threshold $\beta$ used in concept-correlation nicher (CCorrN) on niche purity, impurity, and size in dSprites($\lambda = 0$). Across all approaches, increasing $\beta$ decreases niche purity and niche size, while increasing niche impurity. When $\beta$ is high, only very few concepts can pass the threshold required to be identified as concept nicher for a label. As a result, niche sizes get smaller, which translates to bigger niche complement and, subsequently, more impurity.

Figure 6 allows a qualitative comparison of concept representations. CBM provides the most robust representation (niching scores are less sensitive to $\beta$), while VAEs provide the least stable set of concepts. CCD representations are almost as stable as CBM ones, while CW is a bit closer to VAEs.

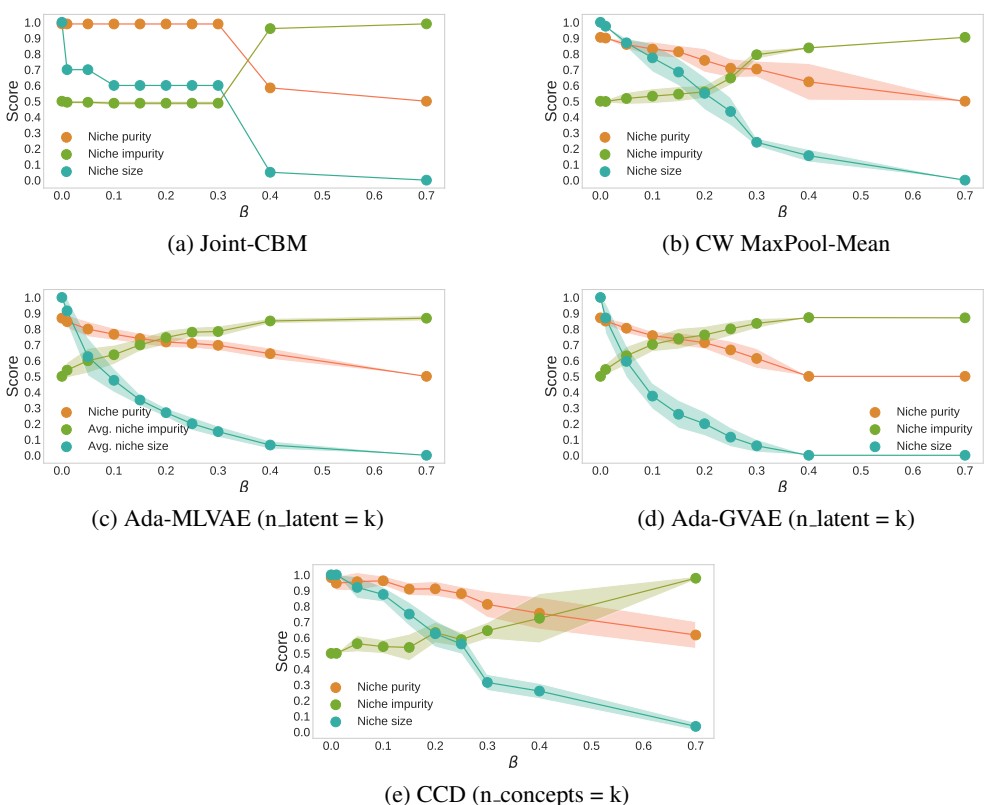

(a) Joint-CBM

(b) CW MaxPool-Mean

(c) Ada-MLVAE (n_latent = k)

(d) Ada-GVAE (n_latent = k)

(e) CCD (n_concepts = k)

Figure 6: The impact of CCorrN's $\beta$ on niche purity, impurity, and size in dSprites($\lambda = 0$).

To compare different models across different data sets, we select a fixed value for $\beta$ for all models ($\beta = 0.2$) in a range where the dependency between $\beta$ and concept scores was approximately linear for most models.

## 6.6 CONCEPT NICHES IN REAL-WORLD DATA

The niching-based metrics reveal information about the task separability and minimality of concepts. To motivate why task separability and minimality are important for evaluating concept quality, we show that these properties are often empirically observed in real world data.

The intuition behind concept niching is relatively simple: for the sake of example, say we have a multilabel task with labels "bird" and "car". Using niching, we can isolate neurons encoding only the information about a car (e.g., a wheel), only the information about a bird (e.g., a feather), and only the information about both of them (e.g., the colour blue). The ideal concept representation has a small niche for each task (one for "car"="wheel", "blue" and one for "bird"="feather", "blue"), simplifying concept-based explanations and downstream procedures.

Figure 7 shows the absolute values of concepts-to-tasks linear correlation coefficients in the Caltech-UCSD Birds-200-2011 dataset (CUB, (Wah et al., 2011b)), as a representative of real-world datasets. The coefficients are computed on ground-truth concept and task labels. The sparsity of the matrix empirically proves that tasks indeed rely on an often small and non-overlapping set of concepts. Thus, the concept niches for each task do not tend to intersect. If this is preserved by the learnt concept representations, then the Niche Impurity Score (NIS), capturing the amount of undesirable mutual information amongst concepts, should be low, while Niche Purity Score (NPS), capturing the amount of inevitable mutual information amongst concepts, should be high. On the other hand,

simultaneously high NIS and NPS suggests that there may be unnecessary mutual information between concepts that counts as leakage.

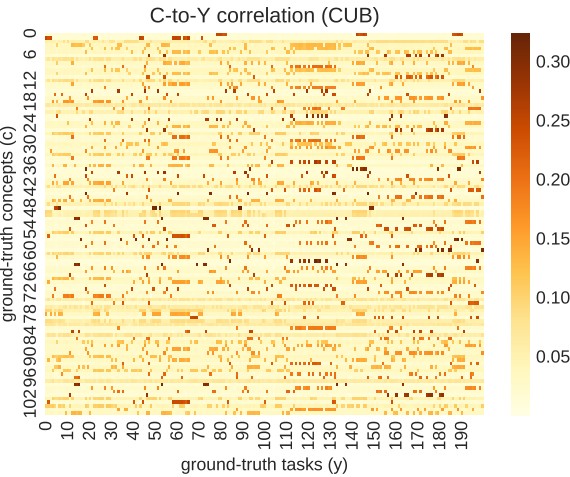

Figure 7: Absolute values of concepts-to-tasks linear correlation coefficients in CUB.

## 6.7 EFFECT OF NETWORK CAPACITY ON ORACLE IMPURITY

We motivate the need of fixing the neural network architecture when contrasting different methods by looking at how the capacity of the selected architecture can affect a method's learnt concept representations. For this, we train a Joint-CBM in the TabularToy($\delta = 0$) dataset (with $\alpha = 0.1$) whose encoder and decoder models are simple ReLU MLPs with two hidden layers. In this experiment, we vary the number of activations in either the encoder and decoder models by setting the number of units in their hidden layers to {capacity, capacity/2} units, while keeping the number of hidden units in their corresponding counterpart fixed to {128, 64}. In this setting, we monitor the predictive accuracy of concept representations w.r.t. their aligned ground truth concepts as well as the representations' oracle impurities. As shown in Figure 8, we observe that as the encoder and decoder capacities decrease, the CBM begins to exhibit significantly higher impurity and lower concept accuracy. Notice that our results also show that the encoder's capacity has a significantly greater effect on the quality of the learnt representations over the decoder's capacity. This motivates the need for a constant architecture with enough capacity to achieve a high concept predictability and serves as a basis for our decision to fix the encoder and decoder architectures across all methods in our evaluation.

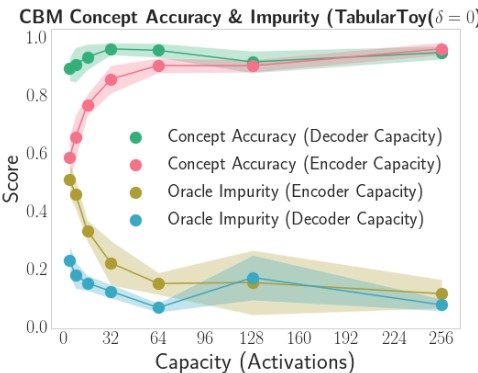

Figure 8: Effect of network capacity (i.e., number of hidden activations used in the encoder and decoder) in a CBM's concept accuracy and oracle impurity.

## 6.8 EXPERIMENT DETAILS

In this section we provide further details on the datasets used for our experiments as well as the architectures and training regimes used to obtain our presented results.

### 6.8.1 DATASETS

For benchmarking, we use three datasets: a simple synthetic toy tabular dataset extending that defined in (Mahinpei et al., 2021), *dSprites* (Matthey et al., 2017), and *3dshapes* (Burgess & Kim, 2018). In order to investigate the impact of dependency between concepts on the quality of concept representation in each method, we allow a varying degree of dependencies in concept annotations and propose the following parameterized tasks:

- *TabularToy*($\delta$): this dataset consists of inputs $\{\mathbf{x}^{(i)} \in \mathbb{R}^7\}_{i=1}^N$ such that the $j$-th coordinate of $\mathbf{x}^{(i)}$ is generated by applying a non-invertible nonlinear function $f_j(z_1^{(i)}, z_2^{(i)}, z_3^{(i)})$ to 3 latent factors $\{z_1^{(i)}, z_2^{(i)}, z_3^{(i)}\}$. These latent factors are sampled from a multivariate normal distribution with zero mean and a covariance matrix with $\delta$ in its off-diagonal elements. The concept annotations for each sample correspond to the binary vector $\mathbf{c}^{(i)} := [\mathbb{1}(z_1^{(i)} > 0), \mathbb{1}(z_2^{(i)} > 0), \mathbb{1}(z_3^{(i)} > 0)]$ and the task we use is to determine whether at least two of the latent variables are positive. In other words, we set $y^{(i)}$ to $\mathbb{1}\big((c_1^{(i)} + c_2^{(i)} + c_3^{(i)}) \geq 2\big)$. The individual functions used to generate each coordinate of $\mathbf{x}^{(i)}$ are the same as those defined in (Mahinpei et al., 2021). As in (Mahinpei et al., 2021), we use a total of $2,000$ generated samples during training and a total of $1,000$ generated samples during testing.

- *dSprites*($\lambda$): we define a task based on the dSprites dataset where each sample $\mathbf{x}^{(i)} \in \{0, 1\}^{64 \times 64 \times 1}$ is a grayscale image containing a square, an ellipse, or a heart with varying positions, scales, and rotations. Each sample $\mathbf{x}^{(i)}$ is procedurally generated from a vector of ground truth factors of variation $\mathbf{z}^{(i)} = [\text{shape} \in \{0, 1, 2\}, \text{scale} \in \{0, \cdots, 5\}, \theta \in \{0, \cdots, 39\}, x \in \{0, \cdots, 31\}, y \in \{0, \cdots, 31\}]$ ($\theta$ indicating an angle of rotation) and is assigned a binary concept annotation vector $\mathbf{c}^{(i)} \in \{0, 1\}^5$ with elements $\mathbf{c}^{(i)} := [\mathbb{1}(z_1^{(i)} < 2), \mathbb{1}(z_2^{(i)} < 3), \mathbb{1}(z_3^{(i)} < 20), \mathbb{1}(z_4^{(i)} < 16), \mathbb{1}(z_5^{(i)} < 16)]$. For this task, we construct a set of 8 labels from the concept annotations by setting $y^{(i)} = \big[c_2^{(i)} c_4^{(i)}\big]_{10}$ if $c_1^{(i)} = 1$ (where we use $[b_1 b_2]_{10}$ to indicate the base-10 representation of a binary number with digits $b_1$ and $b_2$) and $y^{(i)} = 4 + \big[c_3^{(i)} c_5^{(i)}\big]_{10}$ otherwise. Finally, we parameterize this dataset on the dependency number $\lambda \in \{0, 1, \cdots, 4\}$ that indicates the number of random dependencies we introduce across the sample-generating factors of variation (with $\lambda = 0$ implying all factors of variation are independent). For example, if $\lambda = 1$, we introduce a conditional dependency between factor of variations $z_1$ ("shape") and $z_2$ ("scale") by assigning each value of $z_1$ a random subset of values that $z_2$ may take given $z_1$. This subset is sampled by selecting, at random for each possible value of $z_1$, half of all the values that $z_2$ can take. More specifically, if $z_1$ and $z_2$ can take a total of $T_1$ and $T_2$ different values, respectively, then for each $a \in \{0, 1, \cdots, T_1\}$ we constraint $z_2$ to be able to take only $\lfloor T_2/2 \rfloor$ values from the set $z_2 \in \mathcal{Z}_2(a)$ defined as:

$$\mathcal{Z}_2(a) = \begin{cases} SWR(\{0, 1, \cdots, \lfloor \frac{3T_2}{4} \rfloor\}, \lfloor T_2/2 \rfloor) & c_1 = 1 \\ SWR(\{\lfloor \frac{T_2}{4} \rfloor, \cdots, T_2 - 1\}, \lfloor T_2/2 \rfloor) & \text{otherwise} \end{cases}$$

where $SWR(A, n)$ stands for Sample Without Replacement and is a function that takes in a set $A$ and a number $n$ and returns a set of $n$ elements sampled without replacement from $A$. This process is recursively extended for higher values of $\lambda$ by letting the dataset generated with $\lambda = i$ be the same as the dataset generated by $\lambda = i - 1$ with the addition of a new conditional dependency between factor of variations $z_i$ and $z_{(i+1)}$. Finally, in order to maintain a constant dataset cardinality as $\lambda$ varies, we subsample all allowed factor of variations in $\{z_{\lambda+2}, z_{\lambda+3}, \cdots, z_5\}$ by selecting only every other allowed value for each of them. This guarantees that once a conditional dependency is added, the cardinality of the resulting dataset is the same as the previous one. Because of this, all parametric variants of this dataset consist of around $\sim 32,000$ samples.

- *3dshapes*($\lambda$): we define a task based on the 3dshapes dataset where each sample $\mathbf{x}^{(i)} \in \{0, 1, \cdots, 255\}^{64 \times 64 \times 3}$ is a color image containing a sphere, a cube, a capsule, or a cylinder with varying component hues, orientation, and scale. Each sample $\mathbf{x}^{(i)}$ is procedurally generated from a vector of ground truth factors of variation $\mathbf{z}^{(i)} =$ [floor_hue $\in \{0, 1, \cdots, 9\}$, wall_hue $\in \{0, 1, \cdots, 9\}$, object_hue $\in \{0, 1, \cdots, 9\}$, scale $\in \{0, 1, \cdots, 7\}$, shape $\in \{0, 1, 2, 3\}$, orientation $\in \{0, 1, \cdots, 14\}$] and is assigned a binary concept annotation vector $\mathbf{c}^{(i)} \in \{0, 1\}^6$ with elements $\mathbf{c}^{(i)} := [\mathbb{1}(z_1^{(i)} < 5), \mathbb{1}(z_2^{(i)} < 5), \mathbb{1}(z_3^{(i)} < 5), \mathbb{1}(z_4^{(i)} < 4), \mathbb{1}(z_5^{(i)} < 2), \mathbb{1}(z_6^{(i)} < 7)]$. For this task, we construct a set of 12 labels from the concept annotations by setting $y^{(i)} = \left[ c_1^{(i)} c_2^{(i)} c_3^{(i)} \right]_{10}$ if $c_5^{(i)} = 1$ and $y^{(i)} = 8 + \left[ c_4^{(i)} c_6^{(i)} \right]_{10}$ otherwise. As in the dSprites task defined above, we further parameterise this dataset with parameter $\lambda \in \{0, 1, \cdots, 5\}$ to control the number of random conditional dependencies we introduce at construction time. The procedure used to introduce such dependencies is the same as in $dSprites(\lambda)$ but we use $c_5$ rather than $c_1$ for determining the set of values that we sample from. Similarly, we use the same subsampling as in $dSprites(\lambda)$ to maintain a constant-sized dataset, resulting in all parametric variants of this dataset having around $\sim 16,000$ samples.

### 6.8.2 MODEL ARCHITECTURES AND TRAINING

In all of the results we report on the *TabularToy*($\delta$) dataset, for CBM and CCD we use a 4-layer ReLU MLP with activations $[7, 128, 64, 3]$ as the concept encoder $g(\mathbf{x})$ and a 4-layer ReLU MLP with activations $[64, 128, 64, 1]$ as label predictor $f(\hat{\mathbf{c}})$. For CW, we use the same architecture with the exception that a CW module is applied to the output of the concept encoder model. For both CBM and CCD, we train each model for 300 epochs with a batch size of 32. For CW, we train each model for 300 epochs, with CW updates occurring every 20 batches, using a batch size of 128. Finally, for the mixture hyperparameter of the joint loss in CBMs, unless specified otherwise we use a value of $\alpha = 0.1$.

For our CBM and CCD experiments in *dSprites*($\lambda$) and *3dshapes*($\lambda$), we use a Convolutional Neural Network (CNN) with four (Conv + BatchNorm + ReLU + MaxPool) blocks with feature maps $\{8, 16, 32, 64\}$ followed by a three fully-connected layers with activations $\{64, 64, k\}$ for the concept encoder model $g(\mathbf{x})$ (with $k$ being the number of ground truth concepts in the dataset). Furthermore, for the label predictor model $f(\hat{\mathbf{c}})$ we use a simple 4-layer ReLU MLP with activations $\{k, 64, 64, L\}$ (with $L$ being the number of output labels in the task). For CW's concept encoder $g(\mathbf{x})$ we use a CNN with three (Conv + BatchNorm + ReLU + MaxPool) blocks with feature maps $\{8, 16, 32\}$ followed by a (Conv + CW) block with 64 feature maps. For the label predictor, we use a model composed of a MaxPool layer followed by five ReLU fully connected layers with activations $\{64, 64, 64, 64, L\}$. All models evaluated for CBM and CCD are trained for 100 epochs using a batch size of 32. In contrast, CW models are trained for 100 epochs using a batch size of 256 and an CW module update step every 20 batches. Finally a value of $\alpha = 10$ is used during joint CBM training.

For evaluating VAE, $\beta$-VAE ($\beta = 10$), Ada-GVAE and Ada-MLVAE, we use the same architecture as in CBM's and CCD's concept predictor for the encoder and the same architecture as in (Locatello et al., 2020b) for the decoder. The decoder consists of two ReLU fully connected layers with activations $\{256, 512\}$ followed by four ReLU deconvolutional layers with feature maps $\{64, 32, 32, input\_feature\_maps\}$. All DGL models are trained for 100 epochs using a batch size of 32. Weakly supervised models are trained with a dataset consisting of $\frac{2N}{3}$ pairs of images that share at least one factor of variation (with $N$ being the number of samples in the original dataset) while unsupervised models are trained with the same dataset used for CL methods. As in other methods, we train all DGL models using a default Adam optimizer (Kingma & Ba, 2014), with learning rate $10^{-3}$.

When evaluating CCD, we use a threshold of $\beta = 0.0$ for computing concepts scores and the same regulariser parameters $\lambda_1 = 0.1$, $\lambda_2 = 0.1$, $\epsilon = 10^{-5}$ as in Yeh et al. (2020)'s released code for their work in (Yeh et al., 2020). Finally, all CCD models, across all tasks, are trained for 100 epochs and a batch size of 32 using a default Adam optimizer, with learning rate $10^{-3}$.

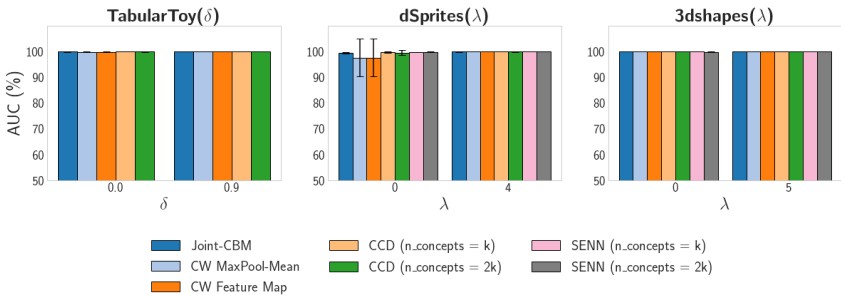

Figure 9: Downstream task predictive AUC for all datasets using original pre-trained models prior to bottleneck construction. Note that because DGL methods have no downstream task supervision in their training pipelines, we do not include those methods.

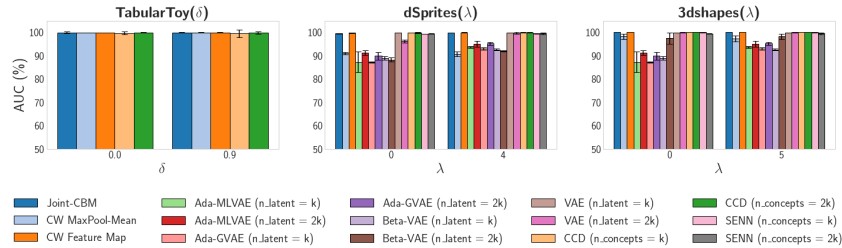

Figure 10: Downstream task predictive AUC for all datasets using the overall set of learnt concepts to predict the task labels. Note that this plot faithfully replicates the downstream task predictive AUC of methods that received direct task supervision in their training (shown in Figure 9).

When benchmarking SENN, we use the same architecture as in DGL methods for the concept encoder $g(\mathbf{x})$ and for its corresponding decoder. Note that the decoder in this case is only used as part of the regularization term during training. For the weight model $\theta(\hat{\mathbf{c}})$ (i.e., the "parameterizer"), we use a simple ReLU MLP with unit sizes $\{\text{input shape}, 64, 64, k\}$ ($k$ being the number of concepts SENN will learn). Finally, we use an additive aggregation function and use $\lambda = 0.1$ as a robustness regularization strength and $\zeta = 2 \times 10^{-5}$ as the sparsity regularization strength, as done in Alvarez-Melis & Jaakkola (2018). We train our SENN models for 100 epochs using a batch size of 32 and a default Adam optimizer with learning rate $10^{-3}$.

## 6.9 DOWNSTREAM TASK PREDICTIVE AUC

Figure 9 shows downstream task predictive AUC for all datasets using raw inputs, in absence of any dependencies as well as the maximum dependencies between concepts. If concept representations from various methods are good surrogates to the inputs, they need to recover the same predictive performance. Figure 10 confirms that this holds to a good degree by looking at the task AUC of a simple ReLU MLP with hidden layers $\{64, 64\}$ trained to predict the corresponding task labels using the concept representations generated by each method.

## 6.10 IMPURITY SCORES (ORACLE AND NON-ORACLE)

Figure 11 confirms that, for 3dShapes($\lambda$), across all models the non-oracle impurity misleadingly shows more impurity that often increases with the dependence of concepts.

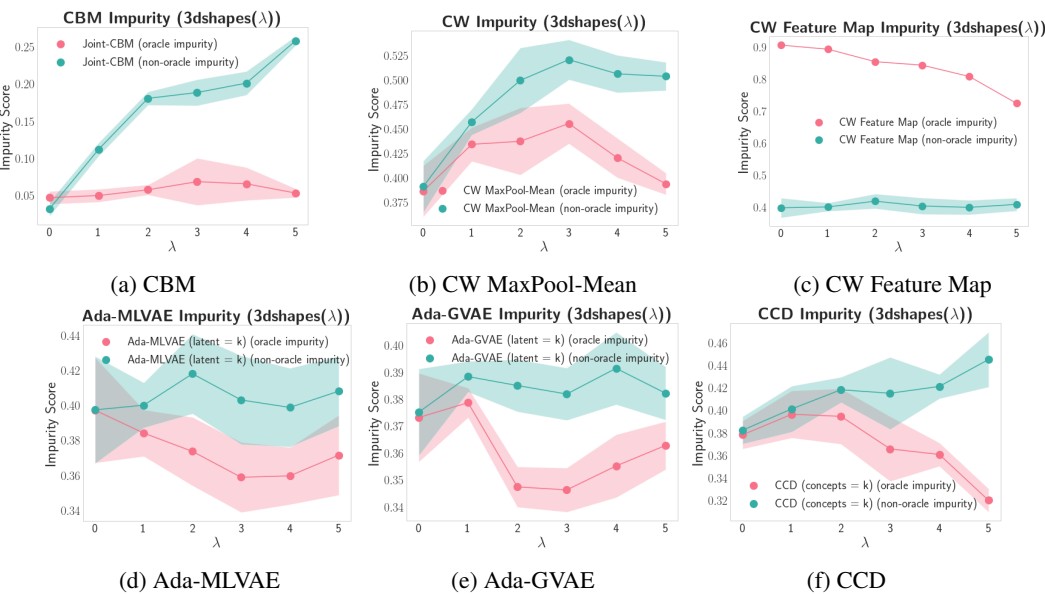

Figure 11: Impurity scores (both oracle and non-oracle) in *3dshapes*($\lambda$).

