# OpenReview forum: "On The Quality Assurance Of Concept-Based Representations"
_ICLR.cc/2022/Conference — ICLR 2022 Submitted_

### Official Review · Reviewer_iQK1 · 2021-10-29

**Correctness:** 3
**Technical Novelty And Significance:** 4
**Empirical Novelty And Significance:** 4
**Recommendation:** 5
**Confidence:** 4

**Main Review:**


Strengths

- metrics which work for correlated underlying factors are really valuable and I think a big blind spot in this space, glad to see this is a focus of this paper
- proposed metrics have some nice underlying intuitions and seem sensible to me
- experiments seem quite extensive, comparing many prior methods from the literature in both CL and DGL

Weaknesses

- paper could greatly benefit from improved clarity throughout: there are a number of points where I struggle to understand exactly what is being communicated
- In particular, clearer descriptions of methods and how those differences manifest in the results section would be helpful - I don't have a deep understanding of all the related work discussed and I have a lot of trouble knowing what the takeaways from the results section should be. Some figures which accent the particular points in question in each section could help, or more analysis of the overall concepts in question for each experiment
- it's not entirely clear conceptually how we're supposed to think about CL and DGL in this paper. Is the paper intended to unify the methods? Contrast them? I don't fully understand what the authors see as the main differences or similarities, or what they want to communicate to the reader on this front. It would be good to spend more time developing a framework and some common language to describe the two approaches
- I give many more specific notes below

Other

Sec 1:
- the acronym DGL for disentanglement learning is strange - I'd personally prefer something like DEL but up to you
Sec 2:
- a working example at the top of Section 2 explaining what each concept might represent would be very helpful, especially to someone like me who is familiar with the overall literature but not on the bleeding-edge methods
-  not sure why we need both g and \phi in this framework?
- can you clarify the role of the downstream task here? it's never really made clear - are there many tasks? just one? do we expect to know it at training time?
- would be useful to define TCAV
- you state that if a function g contains sufficient statistics to perform well on a downstream task f, then g must be invertible; I don't see why this is true, since g may reduce the input \phi to just the sufficient statistics, and much of \phi may not be recoverable
- you list the operating assumption of DGL, what is the analogous assumption for CL?
- you focus on weakly supervised disentanglement, why not weakly supervised CL as well?
Sec 3:
- some sloppiness in language at the top - I'm not sure what the "quality of their learnt concepts" is meant to refer to: only CL (since you used the word "concept")? or both, since you refer to all the "approaches above". This is where I think it would be useful to clarify a particular framework for how you relate these two literatures
- may be beyond the scope of the work, but I agree that the no-leakage assumption is unrealistic - however there are some cases where we want it (the two concepts are truly related) and some cases where we don't (if it's a relic of sampling bias). Just an interesting thought, not sure if there are ramifications in your paper
- it would be good to discuss the CUB example in the main body since it is an important point for the rest of the paper
Sec 3.1
- Def. 1: what does it mean for two representations to be "aligned element-wise"?
- Def. 1: the argmax notation loses me - is the idea that the concept should be categorical? I don't think this was specified earlier
- I don't understand the usage of AUC here - are you assuming binary concepts?
- Implementation concern which should probably be discussed: your estimate of \pi depends on calculating a \sup over functions \psi - this will probably be only approximate, and if this is a related function class (or less powerful) than the encoder in question g, then you might systematically bias your estimate of \pi and OIS
Sec 3.2
- I'm not totally sure the analogy with evolutionary biology is necessary here
- there's an assumption implicit in niching which is that all tasks you care about will necessarily rely on a specific subset of individually predictive dimensions - worth motivating this assumption, it's a little bit different from the standard DGL assumptions
- Def. 6: technically N_j is not an input to NPS, just clean up formalization here to clarify what you mean by "the NPS of \not N_j (v_f)"
Sec 4.2
- "in order to have datasets compatible with both CL + DGL ... construct datasets full described by ground truth gen. factors" - I'm not sure what this means, can't you use any dataset with either of these methods?
Sec 4.3
- clarify exactly what the metric "Mean Concept AUC" refers to - I'm not quite sure how it's calculated
- why is the non-oracle impurity baseline matrix using 1/2 on its off-diagonals - wouldn't the baseline be 0 on the off-diagonals to mimic the situation where we assume no leakage?
- you say that the non-oracle impurity "misleadingly" shows more impurity as dependency increases. However, it's not clear this is misleading, it's possible that the methods are truly doing worse on that data (even relative to oracle). It would be nice to have something external to tell us which is the case
- suggestion: indicate in figures which methods are supervised, this will help interpreting them a lot
- the last sentence of the "oracle impurity demonstrates" paragraph seems to contradict the first: the first says that explicit supervision "encodes purer individual concepts", the last says explicit supervision "does not translate to purer concepts"  - I'm confused which should be true
- the reference to "concept loudness" should be explained to those not familiar with the literature
- the point at the bottom about robustness to concept inter-dependencies seems really important - I would love to see this fleshed out more
Sec 5.
- (iii) you discuss the "weak supervision provided by DGL methods" but aren't there supervised, weakly, and un-supervised DGL methods?
- why should we prefer niching-based metrics?
- this point about efficiency goes over my head, would be good to have a quick discussion in the paper



**Summary Of The Paper:**

The authors propose new metrics for methods in disentanglement learning and concept learning,  which have some nice properties, including robustness to correlations in the underlying factors, and experimentally probe the properties of these metrics on a number of different proposed methods in the disentanglemnt/concept learning literatures.

**Summary Of The Review:**

The authors propose metrics which address a very important problem in the relevant literatures (in particular, underlying factors which may be correlated), but the communication of the ideas and experimental results are not currently clear enough for me to accept the paper as is.

---

> ### Author Response · Authors · 2021-11-19
> **Reply to Reviewer iQK1 - Part 1/3**
>
> Response: We thank the reviewer for their constructive feedback. There are several comments about simplifying definitions, clarification, and fleshing out more details, which we are factoring in while preparing the revised version of the manuscript. Below we provide replies to more substantial comments.
>
> #### **CL and DGL**
> The paper considers two broad paradigms of CL and DGL that aim at generating intermediate representation of the data. It first (i) unifies their formal languages and then (ii) proposes metrics to contrast the quality of intermediate representation within and between the two paradigms.
>
> **$g$ and $\phi$**
> Concepts are often extracted from a hidden layer of the neural network instead of raw input. $\phi$ represents the function that  transforms the raw input into such a hidden representation. If $g$ is applied to the raw input directly, $\phi$ is an identity function.
>
> #### **The role of tasks**
> The intermediate representation of the data we are focusing on in this paper can be extracted from an unsupervised setting (e.g., autoencoders) or a supervised one where essentially a classification problem with task labels (i.e., classification labels) is the starting point.
>
> #### **$g$ must be invertible**
> There may have been a misunderstanding related to this. There is no claim in the paper regarding invertibility of $g$. Can you please explain the question further?
>
> #### **Assumption for CL**
> There is not really an equivalent assumption to that of DGL in CL. However, one may say that CL assumes that the task in hand can be fully explained using units of information that are more abstract and simple than the raw input features.
>
> #### **Weakly supervised CL**
> We are not aware of any weakly supervised CL given that the field is relatively recent. Developing such a method may be a fruitful direction for future work.
>
> #### **quality of their learnt concepts**
> In the background section, second paragraph, we state that for simplicity we refer to the representation found in both DGL and CL  as concepts. We agree with the reviewer that this point needs to be stated clearly earlier in the Introduction and will reflect on that in the revised version of the manuscript.
>
> #### **No-leakage assumption**
> Leakage refers to cases where learnt concepts encode information beyond the ground truth ones they are aligned with. If two ground truth concepts are correlated, then their learnt versions may indeed encode some information about one another. However, such overlap is not referred to as leakage as it is the reflection of the real-world conditions, as pointed out by the reviewer. The OIS metric proposed indeed caters for such cases by factoring in the correlation between ground truth concepts. Encoding information beyond the correlation seen in the ground truth ones counts as leakage and not otherwise.
>
> #### **Assumption implicit in niching**
> Niching metrics work even if all tasks depend on all concepts. Task-separability of concepts is an empirical observation rather than an assumption. We motivate this by showing this phenomenon in CUB, as a representative of real-world datasets. The figure here (https://i.imgur.com/GmJhj61.png) shows the absolute values of concepts-to-tasks linear correlation coefficients in CUB. As evident, tasks (i.e., individual class labels) indeed often rely on a non-overlapping and relatively small set of concepts. We will highlight this motivation in the revised version of the manuscript.
>
> #### **Datasets compatible with both CL + DGL**
> DGL is an unsupervised paradigm for which the notation of a downstream task is non-existent, whereas CL is a paradigm that deals with concepts in a classification task, with or without access to concept labels. Standard DGL datasets are not readily usable in CL, because of a lack of task labels. Standard CL datasets are not readily usable in DGL, as there is no guarantee that concepts are independent as assumed by data generation in DGL, where the data is generated based on an independent set of factors of variations.
>
> #### **Mean Concept AUC**
> Mean concept AUC is calculated by averaging over all the AUCs of predicting individual concepts (which are binary in our tasks) using only the overall set of learnt concepts.
>
> #### **Non-oracle impurity  baseline**
> AUC of 1/2 is representative of random guessing. In off-diagonals, where a concept representation is predicting a concept which they are not aligned with, random guessing shows that the representation can predict the ground truth concepts no better than random, and therefore, there is no leakage.
>
> #### **The last sentence of the "oracle impurity demonstrates**
> We agree that some rephrasing and clarification is required here - thanks for pointing out. Explicit supervision does encode purer concepts, with the exception of CW feature-map, as reported by others cited. We will clarify this exception in the revised version of the manuscript.

---

> > ### Author Response · Authors · 2021-11-19
> > **Reply to Reviewer iQK1 - Part 2/3**
> >
> > **Points related to Definition 1**
> > 1. Aligned component-wise:
> > Intuitively, saying that two concept representations, say $\hat{c}_1$ and $\hat{c}_2$, are aligned component-wise is just saying that the i-th entry in $\hat{c}_1$ encodes for the same concept as the i-th entry of $\hat{c}_2$. The alignment between learnt concepts and ground truth ones is explained in the following extract from the paper. Once such alignment is made, the order of learnt concepts is adjusted to match that of the ground truth ones they are aligned with. Thus there will be a component-wise alignment between the learnt and ground truth concepts.
> > “The alignment between the learnt and ground truth concepts is clear in CBM and CW. In CCD and DGL, where there is no alignment, we compute a greedy alignment between the learnt and ground truth concepts based on the predictive AUC of using a learnt concept to predict each ground truth concept (i.e., a ground truth concept is assumed to be represented by the learnt one that can predict it best). ”
> >
> > 2. argmax notation:
> > This is correct and we appreciate that you pointed this out to us. We have updated our manuscript to make it clear that our metrics, as currently formulated, are applicable only when the concept-to-label model outputs a categorical distribution.
> >
> > 3. The usage of AUC:
> > We use AUC to measure the probability that one is able to correctly predict the j-th ground truth concept using the i-th learnt concept representation when we train a model for this particular task. Although this is technically speaking defined only for binary concepts (and so is most of the concept-explainability literature), note that it can easily be generalised to categorical concepts by either (1) computing the mean one-vs-all AUC across all possible labels a concept may take or, if one has a choice in the training and design of the model which will be evaluated, (2) binarising all concepts so that each label a concept may take is assigned its own binary concept representation.
> >
> > 4. Implementation concerns:
> > It is true that our estimation of $\pi$, which limits the function class of \psi to a one-layered-MLP, can be biased due to our choice of class functions. Nevertheless, we opted to proceed this way for two main reasons: (1) efficiency, as it is computationally intractable to explore the set of all functions $\psi$ with that domain and codomain, and (2)  we are only interested in measuring the divergence between the $\pi$ matrix and the oracle impurity matrix. Because in practice both $\pi$ and the oracle impurity matrix are computed using a search over the same constrained set of functions $\psi$, we expect that any inductive bias will appear in both computations. This means that when measuring the divergence between the two of them, which is what we are interested in measuring for the OIS, this bias should not have a significant effect on the score itself. Finally, note that there is precedence for such a constraint in the family of functions used for similar metrics as seen in [4].
> >
> > **Non-oracle impurity "misleadingly" shows more impurity**
> > For each method in Figure 3, the red colour shows the divergence of learnt concepts from ground truth ones, as the dependency between concepts increases. The blue colour shows the divergence of learnt concepts from ground truth ones, assuming that the ground truth ones were independent, despite the increasing independence between concepts. While comparing red/blue across models allows comparing how well models do comparatively, the point here is to compare the red and blue within each model, where the gap shows that, if the non-oracle impurity is used to evaluate the quality of a concept representation, then all models misleadingly show much less leakage than encoded and recorded by the oracle-impurity. This extra leakage is sourced from the fact that the assumption that all concepts are independent is broken for higher values of $\lambda$ rather than from changes in the representations themselves.
> >
> > **Why should we prefer niching-based metrics**
> > If this is intended to be read as why should we prefer niching-based metrics to oracle impurity, we outline in the second paragraph of Section 5 that when ensuring the overall quality of concept representation is sufficient, niching-based metrics should be favoured to oracle-impurity, due to computational efficiency when one has access to both concept and label annotations. It can be computationally expensive to compute the impurity matrix $\pi$ (and its corresponding oracle matrix) whereas one can quickly compute the linear correlation coefficients needed to evaluate our instantiation of the correlation-nicher. We will update our manuscript to clearly highlight where the efficiency of using our niching metrics comes from.

---

> > > ### Author Response · Authors · 2021-11-19
> > > **Reply to Reviewer iQK1 - Part 3/3**
> > >
> > > **weak supervision provided by DGL methods**
> > > We are not aware of any supervised DGL. It would be great if some references are provided. Unsupervised DGL methods were avoided on the basis of the theoretical impossibility of learning disentangled representations in an unsupervised manner [1], as pointed out in the last paragraph of Section 2, hence using weakly supervised DGLs only. For completeness sake, we have now added benchmarks on unsupervised DGLs too (both vanilla VAE [2] and $\beta$-VAE [3]), as well as for another unsupervised CL method (SENN [5]). We will update our manuscript accordingly with these new benchmarks. See figures (Oracle Impurity Score: https://i.imgur.com/BHbLULq.png; Niche Impurity Score: https://i.imgur.com/6AkkX1a.png, Niche Purity Score:  https://i.imgur.com/tFC8URI.png) for updated plots for both of our metrics. Interestingly enough, in absence of dependencies, unsupervised DGL approaches encode less impurity than the weakly supervised ones. However, when the dependency is high $\beta$-VAE, in particular, tends to struggle more than the semi-supervised DGLs, which is not surprising as the approach is evolved around disentangling factors of variations that indeed have a lot of entanglement.
> > >
> > > [1] Francesco  Locatello,   Stefan  Bauer,   Mario  Lucic,   Gunnar  Ratsch,   Sylvain  Gelly,   Bernhard Scholkopf, and Olivier Bachem. Challenging common assumptions in the unsupervised learning of disentangled representations.  In International Conference on Machine Learning (ICML), volume 97 of Proceedings of Machine Learning Research, pp. 4114–4124. PMLR, 2019.
> > >
> > > [2] Diederik P. Kingma and Max Welling.   Auto-encoding variational bayes.   In Yoshua Bengio and Yann LeCun (eds.), International Conference on Learning Representations (ICLR), 2014.
> > >
> > > [3] Irina Higgins,  Loıc Matthey,  Arka Pal,  Christopher Burgess,  Xavier Glorot,  Matthew Botvinick, Shakir  Mohamed,  and  Alexander  Lerchner.   beta-vae:  Learning  basic  visual  concepts  with  a constrained  variational  framework. In International  Conference  on  Learning  Representations (ICLR). OpenReview.net, 2017
> > >
> > > [4] Chih-Kuan Yeh, Been Kim, Sercan Omer Arik, Chun-Liang Li, Tomas Pfister, and Pradeep Raviku-mar. On completeness-aware concept-based explanations in deep neural networks. InNeuralInformation Processing Systems (NeurIPS), 2020.
> > >
> > > [5] David Alvarez-Melis and Tommi S. Jaakkola.  Towards robust interpretability with self-explaining neural networks.  In Advances in Neural Information Processing Systems (NeurIPS), pp. 7786–7795, 2018.

---

> > ### Comment · Reviewer_iQK1 · 2021-11-23
> > **Response**
> >
> > Thanks for your rebuttal - a couple points to clarify:
> >
> > - invertibility of g: on p3, you say "if this is the case, then there must exist a mapping \psi that recovers \phi(x) from g(\phi(x))". Without any further qualifier, this looks like it is saying that \psi = g^-1 must exist - am I missing something?
> >
> > - language around concepts: in the 2nd paragraph of background, I don't see any language stating that you use the word "concept" to refer to both CL and DGL; rather there is discussion of the common mathematical notation which will be used. I'm referring specifically to the terminology (in natural language) which needs a little more care.
> >
> > - alignment: since it is used in a proof, this concept could use a little more formalization - in my view it's not really a property of the representation so much as its a property of how we interpret it, but I could be misinterpreting the concept.
> >
> > - non-oracle impurity "misleading": I think it's a reasonable demonstration that the metrics are different in this setting, but I think it's important to state there are two reasons why the impurity score could rise in the high \lambda setting: 1. the metric's assumption about leakage, 2. the model might actually be doing a worse job of modelling that data (even accounting for leakage - for instance, high leakage data may be harder to model). It's not clear how we disentangle 1 from 2. This may be difficult to show/beyond the scope of the paper but some more care in discussion would be helpful.

---

> > > ### Author Response · Authors · 2021-11-25
> > > **Replies to iQK1's Rebuttal Comments - Part 1/2**
> > >
> > > We thank the reviewer for going over the new version of our manuscript. Below are our replies to the concerns raised in the comment above:
> > >
> > > ### Invertibility of $g$
> > >
> > > Thank you for further clarifying this concern. In the case of CCD, whose design and rationale we describe in that paragraph, we intended to say that if concept representations $g(\phi(\mathbf{x}))$ capture a complete set of statistics for the task in hand, then one should be able to recover all of the **task-specific crucial information** in $\phi(\mathbf{x})$ from $g(\phi(\mathbf{x}))$ using some function $\psi(\cdot)$. This would enable one to approximate $f(\phi(\mathbf{x}))$ as $f(\psi(g(\phi(\mathbf{x}))))$. This same logic is outlined in [4] (from references below) page 3 above Definition 3.1. While learning $\psi = g^{-1}$ would be ideal, we understand that $\psi$ may instead need to recover only certain aspects of $\phi(\mathbf{x})$ for it to perform equally as well in the task (so there is no strict need to learn an inverse mapping). In fact, due to $g$’s thresholding mechanism, it is not an invertible function by construction so $\psi$ will very likely not be equal to $g$'s inverse. We now understand how our statement in that paragraph can lead to confusion regarding the invertibility of $g(\cdot)$ in CCD, and will update that line accordingly.
> > >
> > > ### Language Around Concepts
> > >
> > > For the sake of simplicity, throughout our paper we refer to both concept representations and latent codes as “concepts”. Based on your previous feedback, we updated our introduction to state that “We unify the language and notation across CL and DGL by framing factors of variation and latent codes in DGL as ground truth concepts and concept representations in CL, respectively.” We then mathematically represent this use of language when describing our mathematical notation in the 2nd paragraph of our background section. To add further clarity, we will **reiterate** our use of “concepts” for latent codes in the background section.
> > >
> > > ### Alignment
> > >
> > > We completely agree that an alignment is not a property of the representation but rather a semantical assignment to each axis of the representation. At its most formal, and as described in Appendix 6.4, an alignment $\mathcal{A}:$ {$1, \cdots, k$} $\mapsto$ {$1, \cdots, k^\prime$} between a concept representation $\mathbf{\hat{c}} \in \mathbb{R}^{d \times k^\prime}$ and a ground truth concept vector $\mathbf{c} \in \mathbb{R}^{k}$ is an **injective** function such that $\mathcal{A}(i) = j$ indicates that concept representation $\mathbf{\hat{c}}_{(:, j)}$ is *semantically* aligned with ground truth concept $c_i$. In other words, $\mathbf{\hat{c}}$ encodes information that is specific to ground truth concept  $c_i$. Notice that when we generate as many concept representations as ground-truth concepts (e.g., as in CBM and CW) we will have $k^\prime = k$. For the sake of simplicity, and without loss of generality, in our definitions we simply assume that representations are *aligned element-wise* and describe it informally as “for all $l \in$ {$1, \cdots, k$} , the $l$-th concept representation of $\mathbf{\hat{c}}^{(i)}$ encodes for the same concept as the $l$-th concept label in $\mathbf{c}^{(i)}$” (in Definition 1). This can be formally thought of as assuming that $\mathcal{A}(l) = l$ for all $l \in$ {$1, \cdots, k$}. We opted for the informal definition rather than the formal one to simplify and make our definitions more clear. Nevertheless, we will point out in our paper’s main body that a formal definition of an alignment can be found in Appendix 6.4 and extend that appendix to formally clarify the terminology we use.

---

> > > > ### Author Response · Authors · 2021-11-25
> > > > **Replies to iQK1's Rebuttal Comments - Part 2/2**
> > > >
> > > > ### Non-oracle Impurity “Misleading”
> > > >
> > > > As mentioned, there is certainly a difficulty disentangling the possible causes of the discrepancy shown in Figure 2 between oracle and non-oracle impurity. For the sake of an example, however, consider our results shown in Figure 3 for CBM: we see that our impurity metric remains relatively stable as the number of dependencies changes in the data while the non-oracle impurity increases as the amount of dependencies is introduced. Nevertheless, a closer look at Figures 2, 9, and 10 shows that CBMs learn concept representations that have equally as good predictive power (almost 100%) both when $\lambda = 0$ (i.e., no dependencies) and $\lambda = 5$ (maximum number of dependencies) in 3dshapes$(\lambda)$. In fact, when looking at how the predictive performance of CBM’s representations change as we vary $\lambda$, we see that they remain very stable, with both being almost 100% (see here: https://i.imgur.com/ZlIaJ5B.png). This indicates that the cause of the increase in the non-oracle impurity is very likely linked to the incapability of the non-oracle impurity metric to correctly generalize to scenarios where concepts are correlated. A similar effect is observed in other models and datasets. To clarify this point, we will include a brief description highlighting the discrepancy between what we see in the non-oracle impurity in Figure 2 and the predictive performance stability we see in Figures 2, 9, and 10.
> > > >
> > > >
> > > > Many thanks for going through our updated manuscript. Your comments have helped improve the quality of our work. We are happy to answer any additional questions or provide other details that would help support our paper’s acceptance.

---

### Official Review · Reviewer_yfkD · 2021-11-05

**Correctness:** 4
**Technical Novelty And Significance:** 2
**Empirical Novelty And Significance:** 3
**Recommendation:** 5
**Confidence:** 4

**Main Review:**

Concept-based explanations are a relatively new idea; the first papers on this topic, including TCAV and its extensions, were post-hoc methods that did not require specialized model architectures. More recently, the concept bottleneck idea suggested training a model that explicitly encodes concepts at an intermediate layer (enabling interventions, etc). In parallel, as the authors say, there has been a rich literature on unsupervised/semi-supervised disentangled representation learning, where models are trained that may or may not capture semantically meaningful, decorrelated concepts in their latent spaces. It is natural to wonder to what extent these various methods can result in representations that are useful from a concept-based model explanation perspective.

Before getting into the authors' contributions, it's probably worth saying that the intuitive answer here (for me, at least) is that concept supervision is key, and without it we can't expect a model to learn a latent space that is well-aligned with concepts that we don't tell it about. Ultimately, this appears to be the takeaway from the experiments (see figures 2, 4, 5), but it's not a particularly exciting or surprising result; it's showing what people would expect to begin with, but with a little bit of rigor.

Below, I'll summarize and comment on the authors' two proposed metrics.

### Oracle impurity score

The proposed OIS score basically assesses to what extent the learned concepts predict the ground truth concepts equivalently to how the ground truth concepts would. Some previous work suggests that concepts should not be predictive of one another because this would constitute "leakage," which is problematic for experiments with concept interventions. To remedy that, because concepts in fact are related (this can likely be seen in the ground truth annotations for most datasets), the authors consider how well each ground truth concept can predict another (using AUC, for example) and then ask whether the learned concept achieves the same predictive performance.

More specifically, the authors build a matrix of predictive accuracies for all pairs of concepts, using both the ground truth and learned concepts as predictors. They then take the Frobenius norm of the difference between these two matrices and apply a couple constant factors.

This is fine, it seems like an improvement on previous metrics for measuring leakage. It also seems related to the idea of measuring how well the learned concepts can predict the ground truth concepts in a 1-1 manner, which is one of the main metrics in the CBM paper. While that metric effectively assesses the mutual information between matched pairs of learned and ground truth concepts while ignoring inter-concept relationships, this one measures (roughly) the delta in mutual information across all pairs.

A couple questions and concerns about this:
- The bounds used to guarantee that the score is in [0, 1] seem to require the use of AUC as a performance metric. What if we want to use something else, e.g. because the concept is multivariate or real-valued?
- If you want to quantify the mutual information for a discrete concept, why not use cross entropy loss? This would approximate $H(c_i | c_j)$ and $H(c_i | \hat c_j)$, whose difference is equal to $I(c_i | c_j) - I(c_i | \hat c_j)$. This seems more aligned with what you're talking about than AUC.
- The use of a Frobenius norm on the difference in matrices of prediction accuracies seems a bit heuristic. It's fine, it just seems a bit arbitrary and I wonder if there's a better way to do this depending on what the particular accuracy metric is (e.g., if we had exact mutual information values).
- The notation in definition 1 is a pretty odd way of saying you train a model $\psi_j$ to predict one concept from another. I would not have been able to understand this without the explanatory text below. Please consider rewriting this a different way.

### Niching scores

The next metric tries to provide some measure of concept-based representation quality without requiring access to ground truth concepts. This is quite a leap, because it requires resorting to some notion of quality that departs from how previous work has assessed concept-based representations.

The authors suggest that for a good concept-based representation, we should be able to identify a small number of concepts that are predictive of each output dimension, and that the remaining concepts should not be predictive (because this would constitute leakage). I read this part of the paper a couple times and I just don't buy this argument, I don't see how the ability to "niche" concepts into groups that are useful/useless for each output dimension is a measure of the representation's quality. This is not what I would think of as a measure of whether my learned concepts capture what they're supposed to capture. And the ability to do well on these scores seems dataset-dependent: what if there are output dimensions where every concept is useful, or where no concept is useful? It's a nice idea to try to make a metric that doesn't require access to ground truth concepts, but I don't quite agree with the premise of this particular approach. (Though I'm curious to hear if other reviewers disagree.)

Some other questions and concerns:
- I'm not so sure about framing the fact that previous metrics require access to ground truth concepts as a weakness. First off, if you're using some of these methods (e.g., CBM) we know you have access to ground truth concepts, so it's not an unrealistic assumption. Second, existing notions of high-quality concept-based representations are defined specifically based on learned representations capturing known concepts, so access to those concepts is clearly necessary; you haven't gotten around that need, you've just opted for an orthogonal notion of representation quality.
- Isn't a high NPS and low NIS virtually guaranteed by the construction of your concept niche? You're effectively doing feature selection to ensure that the predictive concepts are in one group and the useless ones are in another (for a given output dimension). For any representation at all, it should not be surprising if NPS is high and NIS is low. Or am I missing something here?

As explained above, I have some qualms about this metric, but given its current state here are a couple other comments:
- When defining your "concept nicher" (i.e., a way to decide which concepts are relevant to a given output dimension), there are a couple problems with your use of correlation. First off, it's worth noticing that correlation is essentially a measure of the MSE from a fitted linear regression model (check the math on this for the specific relationship). With that in mind, there's a better option than taking the max score when you have multivariate learned concepts: derive the MSE from a fitted multivariate linear model that takes in all the concept dimensions. But why not allow a nonlinear model? Also, what if the label is not real-valued but a classification label? Rather than using correlation, wouldn't it be better to fit a classifier and check the cross entropy loss? Yes, this requires fitting a model, but that step is only required once per method evaluation.
- How are we supposed to choose beta? Different beta values will result in very different outcomes. For example, if I set beta close to 0, I expect we'll see great NPS/NIS scores (because everything will be deemed relevant to every output dim).
- Overall, there's a lot of wiggle room in your definition of "concept niching" and it would be nice if there were a better justified default option. On the other hand, this aspect of your paper is essentially feature selection, and that's a hard problem with approximations of varying quality and computational cost.

### Experiments

The experiments section uses the proposed metrics to compare some SOTA concept-based and DGL algorithms. As mentioned at the beginning of my review, these experiments overall felt a bit too much like sanity checks (particularly figure 3), because the outcome was mostly what I would have expected to begin with. Also, it could have been nice to use some non-toy datasets.

### Other comments

- While you're at it evaluating various methods, why not include any unsupervised (rather than semi-supervised) DGL algorithms? It would be interesting to see how much worse these do than the other approaches (which is perhaps to be expected because of the works cited).
- The OIS score appears to require a known alignment between learned and ground truth concepts, is that right? Am I correct in understanding that this is somehow known for semi-supervised DGL approaches but not unsupervised ones?
- The definitions in this paper are not easy to read. They're too verbose and too long, please try to make them more concise.


**Summary Of The Paper:**

The authors consider the question of whether recent concept-based learning algorithms, as well disentangled representation learning algorithms, result in high-quality representations. In particular, they consider what high-quality should mean in terms of the relationship with ground truth concepts and the ability to make accurate predictions for a downstream task. To this end, they propose two main metrics for representations that are explicitly or implicitly encouraged to encode concepts: 1) a score that captures how well the learned representation preserves the relationships between concepts (which may be correlated), and 2) a score that captures how well concepts can be split into groups that are useful/useless for predicting particular label dimensions.

**Summary Of The Review:**

It's a good idea to assess whether unsupervised and semi-supervised DGL algorithms can learn high-quality concept-based representations, and in doing so the authors developed two metrics that they suggest can be used to compare concept-based representations both with and without access to ground truth concepts. However, the results from their study do not seem very surprising or impactful, and I was not convinced of the rationale behind these metrics (for the reasons described above). For that reason, my rating is that this work is marginally below the acceptance threshold.

---

> ### Author Response · Authors · 2021-11-21
> **Reply to Reviewer yfkD - Part 1/3**
>
> We thank the reviewer for their constructive and insightful comments. The clarity of the definitions will be improved upon and succinctness will be ensured in the revised version of the paper. In the meantime, below we address the main concerns raised in the review:
>
> ## Replies to Oracle Impurity Score Concerns
>
> Thank you for your questions and comments regarding the OIS. We agree with the reviewer that our description of the metric could be more clear and succinct. We will appropriately address this in our updated manuscript.
>
> ### AUC as Proxy for Mutual Information
>
> We first agree that, for OIS to be bounded in [0,1], it requires that AUC is used as a proxy to measure the mutual information between learnt concepts and ground truth concepts. In our work, we focus on this instantiation and mention in the paragraph preceding definition 2 that, while one may choose to measure this in different ways, for the purposes of our evaluation we use AUC as a proxy for mutual information. In our updated manuscript, we aim to motivate this decision based on the following two facts: (1) AUC provides a tractable way of computing a proxy for the mutual information between the two variables of interest and (2) AUC trivially allows our score to be bounded in [0, 1]. While we believe that using something such as cross entropy loss may be interesting and similarly aligned with our goals, we opted away from such approaches mostly due to the difficulty that would come from bounding the resulting metric. We believe, however, that such an approach may be interesting to try as an unbounded version of this metric and is worth exploring as future work.
>
> Similarly, we agree that other similarity metrics could be used to quantify the discrepancy between the oracle matrix and the representation’s purity matrix. We primarily chose the Frobenius norm due to its efficiency and simplicity and will update our manuscript to make this decision’s rationale explicit. Furthermore, notice that the use of the Frobenius norm, together with AUC scores to compute the entries in the matrices, is what enables easy normalization of our norm. This is very useful when attempting to interpret the output score.
>
> Finally, as correctly pointed out by the reviewer, the current instantiation depends on concepts being binary. Nevertheless, notice that, as it is the case for most of the literature in CL ([6], [7], [2]), in this work we are concerned with tasks in which ground truth concepts are binary. This, however, does not imply that the OIS could not be applied to multivariate concepts. One could, for example, compute the mean one-vs-all AUC across all possible labels a concept may take or, if one has a choice in the training and design of the model which will be evaluated, binarize all concepts so that each label a concept may take is assigned its own binary concept representation. Nevertheless, notice that this does imply that our metrics are not applicable for real-valued concepts. We believe that this can be justified by noticing that there has not been any substantial work on learning concept representations where ground truth concepts are real-valued. To clarify this, we are adding a line in our updated manuscript where we make this limitation explicit and state that our work is primarily focused on binary concepts and multivariate tasks (for niching scores).

---

> > ### Author Response · Authors · 2021-11-21
> > **Reply to Reviewer yfkD - Part 2/3**
> >
> > ## Replies to Niching Scores Concerns
> > ### Assuming access to ground truth concepts
> > Assuming access to ground truth concepts may not be a weakness but it is definitely a limitation when learning concepts in a not fully supervised manner. Metrics that make such assumptions are fully compatible with approaches such as CBM and CW that benefit from concept supervision, however, to be used in other paradigms where concept annotations are not available or are available partially, one still needs to assume access to full concept annotation. It is hard to imagine a real-world case where concept labels are available and used for verification, but not in the learning. Consequently, previous work has attempted to introduce "metrics/quality control mechanisms" that do not rely on ground truth concepts. These include importance score in [3] and completeness score in [4].
> >
> > ### High NPS and low NIS
> > High NPS is not necessarily accompanied by low NIS. In cases where every learnt concept representation encodes information about nearly all ground truth ones, both NPS and NIS tend to be high (i.e., when a concept representation is able to predict with high probability both aligned and misaligned concepts). We observe this in CW, especially when using the entire concept maps as concept representations, as well as in Self-Explaining Neural Networks (SENN) [5], an unsupervised concept learning method we will include as part of our updated manuscript. You can see both SENN’s and CW’s NIS in https://i.imgur.com/6AkkX1a.png and their NPS in  https://i.imgur.com/tFC8URI.png. Note that the results we obtain for SENN differ from those obtained for CCD, our other unsupervised CL method. We believe this to be a consequence of CCD’s training process including a regularization term that encourages coherence between concept representations in similar samples and misalignment between concept representations in dissimilar samples.
> >
> > ### Selecting $\beta$
> > One way to select beta, and also to measure the quality of the concept representations, is to compute niching scores for different values of $\beta$ and generate a plot as we have done in Fig. 7. Such a plot highlights how robust your concept representation is and allows you to see whether or not reducing the niche sizes by increasing $\beta$ quickly results in worse niching scores. One can then select $\beta$ by picking a value for which the niche size becomes relatively stable, or at least linear as a function of $\beta$. For example, we selected $\beta = 0.2$ in our evaluation by noticing from our plots in Figure 7 that niche sizes, and niche scores, remain relatively linear and stable around that value. We will update section 6.3 accordingly to include a suggestive strategy for how to select $\beta$.
> >
> > ### Defining Concept Nicher
> > The attempt to give a general definition of a nicher is intentional to leave room for various instantiations. We chose to use a specific instantiation of the nicher using linear correlation coefficients for the sake of simplicity and efficiency. Nevertheless, one can opt for a non-linear model in a different instantiation and is something we would like to explore as part of future research.
> >
> > ## Replies to Other Concerns
> > ### Unsupervised DGL
> > We have now added two unsupervised DGL methods to our experiments, vanilla VAE [3] and $\beta$-VAE [4] (with $\beta = 10$). You can see our updated plot for OIS in https://i.imgur.com/BHbLULq.png while our plots for NIS and NPS can be found at https://i.imgur.com/6AkkX1a.png and https://i.imgur.com/tFC8URI.png, respectively. Interestingly enough, we observe that, in absence of dependencies, unsupervised DGL approaches encode less impurity than the weakly supervised ones. However, when the dependency is high, $\beta$-VAE, in particular, tends to struggle more than the semi-supervised DGLs, which is not surprising as the approach has the underlying assumption that the different factors of variation are independent and incentivizes this separation by the multiplier $\beta$ in its loss function.
> >
> > ### OIS and alignment between learned and ground truth concepts
> >
> > You are correct in your understanding that an alignment is required to compute the OIS. Furthermore, this alignment is not known for the DGL methods we evaluate within our work (both semi-supervised and unsupervised). To address this, as pointed out under the heading “Metrics that assume concept independence may be misleading”, in CCD and DGL (weakly or unsupervised), where there is no alignment, we compute a greedy alignment between the learnt and ground truth concepts based on the predictive AUC of using a learnt concept to predict each ground truth concept (i.e., a ground truth concept is assumed to be represented by the learnt one that can predict it best). We opted for this greedy strategy for efficiency purposes mainly. Nevertheless, future work may explore better alignment mechanisms, as well as fully exploit the whole search space when the number of concepts is small.

---

> > > ### Author Response · Authors · 2021-11-21
> > > **Reply to Reviewer yfkD - Part 3/3**
> > >
> > > ## Replies to Experiment Concerns
> > >
> > > ### Take away from experiments
> > >
> > > Showing that supervision results in better quality of concepts as opposed to no supervision may be expected, but there are definitely exceptions to this: while this intuition is true for CBMs, our results actually show that it does not always hold. CW is surprisingly worse than unsupervised CL methods such as CCD.
> > > In addition, the fact that mere label supervision can generate concept representations that are as high quality as those benefiting from concept supervision is indeed novel and impactful given the high cost of proving concept annotations. The same observation holds for the weak supervision provided in DGL: although such weak supervision was introduced in response to the difficulty of learning representations that capture the underlying factors of variations in the data in an unsupervised manner, empirically it lags far behind supervision from task or concepts. Interestingly enough, the new experiments we have run on unsupervised DGL (thanks for suggesting that), namely vanilla VAE [1] and $\beta$-VAE [2], show that our metrics record that the representations learnt by these models can be at least as good, if not better, than those learnt by weakly supervised DGLs. This results in the representations learnt by unsupervised DGLs being significantly better than the weakly supervised ones at predicting both concepts and task labels (see the following two figures at https://i.imgur.com/jRwZCiL.png and https://i.imgur.com/LFq4D5m.png, respectively).
> > > We agree that the result sections could be more clear on what results are in line with intuition, with our empirical studies confirming those with rigour,  and what results are new and can serve as the basis of making informed choices when choosing a method for finding an intermediate representation of the data. We will accommodate this in the revised version of the manuscript.
> > >
> > > ## References
> > >
> > > [1] Amirata Ghorbani, James Wexler, James Y. Zou, and Been Kim. Towards automatic concept-based explanations. In Neural Information Processing Systems (NeurIPS), pp. 9273–9282, 2019.
> > >
> > > [2] Chih-Kuan Yeh, Been Kim, Sercan ̈Omer Arik, Chun-Liang Li, Tomas Pfister, and Pradeep Raviku-mar. On completeness-aware concept-based explanations in deep neural networks. In NeuralInformation Processing Systems (NeurIPS), 2020.
> > >
> > > [3] Diederik P. Kingma and Max Welling.   Auto-encoding variational bayes.   In Yoshua Bengio and Yann LeCun (eds.), International Conference on Learning Representations (ICLR), 2014.
> > >
> > > [4] Irina Higgins,  Loıc Matthey,  Arka Pal,  Christopher Burgess,  Xavier Glorot,  Matthew Botvinick, Shakir  Mohamed,  and  Alexander  Lerchner.   beta-vae:  Learning  basic  visual  concepts  with  a constrained  variational  framework. In International  Conference  on  Learning  Representations (ICLR). OpenReview.net, 2017.
> > >
> > > [5] David Alvarez-Melis and Tommi S. Jaakkola.  Towards robust interpretability with self-explaining neural networks.  In Advances in Neural Information Processing Systems (NeurIPS), pp. 7786–7795, 2018.
> > >
> > > [6] Koh, Pang Wei, et al. "Concept bottleneck models." International Conference on Machine Learning. PMLR, 2020.
> > > [7] Chen, Zhi, Yijie Bei, and Cynthia Rudin. "Concept whitening for interpretable image recognition." Nature Machine Intelligence 2.12 (2020): 772-782.

---

> > ### Comment · Reviewer_yfkD · 2021-11-26
> > **Thanks for response**
> >
> > Thanks to the authors for their response. A couple thoughts on what they've discussed:
> >
> > **About the choice of AUC.** If the main motivation for using AUC is to bound the metric in [0, 1], that's not a very good reason. The mutual information $I(X; Y)$ is bounded by $0 \leq I(X; Y) \leq H(Y)$, and if $Y$ is discrete then $H(Y)$ is easy to estimate. So you could achieve a similarly simple range of scores for mutual information rather than AUC. Not that AUC is a terrible choice, my complaint here was that it's a bit arbitrary and somewhat difficult to generalize to non-binary concepts.
> >
> > **About the takeaway from experiments.** Thanks for clarifying the more nuanced interpretation of the various methods' performance. Indeed, it's not as simple as concept supervision yielding good representations and everything else failing. A new takeaway seems to be that semi-supervised DGL may not outperform unsupervised DGL (in the sense of having good concept-based representations), which casts some doubt on this line of work.
> >
> > **Other clarifications.** Thanks for clarifying a couple other points. Describing how you chose $\beta$ will be helpful for readers, although what you've described sounds a bit subjective. Adding the additional methods is definitely an improvement. The various limitations of your metrics is helpful to point out, though I mostly agree that there are relatively simple ways to generalize them to real-valued or multi-class concepts.
> >
> > Overall though, I remain only moderately excited about the execution and impact of this paper, so I'm leaving my score as is.

---

### Official Review · Reviewer_7LHy · 2021-11-07

**Correctness:** 4
**Technical Novelty And Significance:** 3
**Empirical Novelty And Significance:** 3
**Recommendation:** 5
**Confidence:** 4

**Main Review:**

This paper introduced metrics to evaluate the concept quality generated by both methods i.e. concept-based representation learning and disentanglement learning in the scinerio of concept supervision and correlation of concepts. For this purpose, mainly three family of methods were considered i.e. concept-based representation learning both with and without concept supervision and semi-supervised disentanglement learning method. Based on this metrics, some recommendations were made regarding requirements of explicit concept supervision for concept-based representation learning methods and weak supervision for disentanglement learning methods.

This paper has below weaknesses/clarifications/suggestions:
1)It's not very clear how the concept property (ii) is related to the contributions made based on concept correlations and availability of ground truth cocnepts. The property is definitely a desirable property for the concepts, but how the proposed metrics capture whether a set of generated concepts follow this property, specially when the ground truth concepts are not available.

2)How each entry of the purity matrix is calculated? Do all the data points are required to calculate each matrix entry? Please elaborate the line "In practice, we parameterise the family of functions ψ j by training, via gradient descent, a ReLU MLP with 32 hidden activations" in this context.

3)A very important work (https://arxiv.org/abs/1806.07538) is missing from the paper, which can be a very good representative of unsupervided concept learning methods. This work has used both label supervision and decoder based network for better concept learning. Extending in the same line, What is your opinion on (possibility of) merging these two types of methods (CL and DGL) for better concept learning? If you think it's possible, then how would the proposed metrics would be helpful here?

4)Figure 8 shows the effect of network capacity. On top of that, it would be interesting to see the individual effects of capacities of encoder and the decoder. To be precise, improving the capacities of both the encoder and the decoder would result low impurity and higher cocnept accuracy, but it's important to check if capacity of the encoder has more effect on concept quality than capacity of the decoder.

**Summary Of The Paper:**

The authors have put decent effort to bring concept-based representation learning and disentanglement learning together under one umbrella in terms of the quality of generated concepts in presence as well as absence of ground truth concept labels. Some related metrics were proposed for evaluation of the quality of concepts for both these methods. Based on these studies, presented in the paper, some important conclusions were made based on requirements of concept supervision and their effects on final concept quality as well as the predictive performance of the model.

**Summary Of The Review:**

This is an interesting work that tried to unify concept-based representation learning and disentanglement learning and proposed corresponding metrics that helped to find some important conclusions. Answering/commenting on the points, that I posted under the main review, can significantly improve the quality of this work.

---

> ### Author Response · Authors · 2021-11-19
> **Reply to Reviewer 7LHy - Part 1/2**
>
> We thank the reviewer for their constructive comments and feedback.
>
>
> ### Concept property (ii)
> Property (ii) states that concepts should preserve the amount of mutual information observed in ground truth concepts or factors of variation. The Oracle impurity score is precisely doing that by measuring the divergence of the learnt concepts from the ground truth ones. We agree that we need to clarify that whilst in absence of access to ground truth concepts, the niching-based metrics reveal information about the task separability and minimality of concepts, they do not fully realise property (ii). This will be reflected in the revised version of the paper.
> In the meantime, to motivate why task separability and minimality are important for evaluating concept quality, below we show that these properties are often empirically observed in real-world data. The figure in https://i.imgur.com/GmJhj61.png, which we will include in our paper, shows the absolute values of concepts-to-tasks linear correlation coefficients in CUB, as a representative of real-world datasets. As evident, tasks indeed rely on an often non-overlapping set of concepts. Thus, the concept niches for each task do not tend to intersect. If this is preserved by the learnt concept representation then the Niche Impurity Score (NIS), capturing the amount of undesirable mutual information amongst concepts, should be low, while Niche Purity Score (NPS), capturing the amount of inevitable mutual information amongst concepts, should be high. On the other hand, simultaneous high NIS and NPS suggest that there may be unnecessary mutual information between concepts that counts as leakage.
>
>
>
> ### Purity matrix entries
> We compute the (i, j)-th entry of the purity matrix as follows: we split the original testing data $(X_\text{test}, Y_\text{test}, C_\text{test})$ into two disjoint sets, a new training set $(X_\text{train}^\prime, Y_\text{train}^\prime, C_\text{train}^\prime)$ and a new testing set $(X_\text{test}^\prime, Y_\text{test}^\prime, C_\text{test}^\prime)$, using a traditional 80%-20% split. We then use the concept representations learnt for the i-th concept for samples in $X_\text{test}^\prime$ to train a ReLU MLP $\psi(\cdot)$ with a single hidden layer with 32 activations to predict to truth value of the j-th ground-truth concept. In other words, we train  $\psi(\cdot)$ using labelled samples {$\Big( g \big( \phi ( \mathbf{x}^{(l)} ) \big)_i, \mathbf{c}_j^{(l)} \Big)   |   \mathbf{x}^{(l)} \in X_\text{train}^\prime \wedge \mathbf{c}^{(l)} \in C_\text{train}^\prime $}. Finally, we set the (i, j)-th entry of the purity matrix as the AUC obtained when evaluating $\psi(\cdot)$ on the new testing set $\big( g \big(\phi(X_\text{test}^\prime) \big), C_\text{test}^\prime)$. Note that we therefore do not require all of the data to compute the entries of this matrix but we rather depend on using the original testing split exclusively. We will address this lack of clarity in the revised version of the manuscript.
>
>
>
> ### Missing related work
>
> We acknowledge the relevance of the missing work (SENN in short) mentioned. Thanks for pointing this out. We have now added SENN as a benchmark and will update the manuscript with this accordingly. See figures (Oracle Impurity Score: https://i.imgur.com/BHbLULq.png; Niche Impurity Score: https://i.imgur.com/6AkkX1a.png, Niche Purity Score:  https://i.imgur.com/tFC8URI.png) for updated plots for both of our metrics which now include SENN results as well as two unsupervised DGLs (both vanilla VAE [1] and $\beta$-VAE ($\beta = 10$) [2]). The Oracle Impurity Score reveals that representations learnt by SENN are considerably more prone to unnecessary leakage (i.e., impurity) than those learnt by CCD, which is the closest approach to it. We observe that this is because SENN’s concept representations are able to predict non-aligned concepts better than those learnt by CCD. We believe that the decrease in leakage in CCD is a consequence of CCD’s training process including a regularization term that encourages coherence between concept representations in similar samples and misalignment between concept representations in dissimilar samples. Niching scores corroborate this hypothesis by showing a similar trend to that observed in oracle impurity.
>
> ### Proposed metrics in various learning regimes
>
> The applicability of metrics depends on access to ground truth concepts in the case of oracle impurity metric and access to task labels in the case of niching-based ones. Thus any learning regime that gives access to either requirement can be assessed based on the proposed metrics, including those that may merge CL and DGL in various ways.

---

> > ### Author Response · Authors · 2021-11-19
> > **Reply to Reviewer 7LHy - Part 2/2**
> >
> > ### Combining CL and DGL for better concept learning
> >
> > In situations where concepts are known to be fully disentangled, using DGL with some type of weak supervision related to concepts or task labels can indeed prove fruitful as a direction for future work. As mentioned in the paper though, it is hard to realistically assume real-world concepts are disentangled.
> >
> >
> > ### Capacity of encoder  vs. decoder
> >
> > Running experiments on increasing the capacity of the encoder and decoder separately confirms that the encoder's capacity has a much more important role in the quality of the representations than that of the decoder as pointed out by the reviewer. This can be seen in the following figure https://i.imgur.com/cRcc4bG.png where the capacity of the encoder/decoder is varied while its counterpart's capacity is left fixed. We will highlight this in the revised version of the manuscript.
> >
> >
> > ### References
> > [1] Diederik P. Kingma and Max Welling.   Auto-encoding variational Bayes.   In Yoshua Bengio and Yann LeCun (eds.), International Conference on Learning Representations (ICLR), 2014.
> >
> >
> > [2] Irina Higgins,  Loıc Matthey,  Arka Pal,  Christopher Burgess,  Xavier Glorot,  Matthew Botvinick, Shakir  Mohamed,  and  Alexander  Lerchner.   beta-vae:  Learning  basic  visual  concepts  with  a constrained  variational  framework. In International  Conference  on  Learning  Representations (ICLR). OpenReview.net, 2017

---

### Author Response · Authors · 2021-11-23
**Summary of Changes in Revised Manuscript**

We would like to thank reviewers for all their feedback, suggestions, and comments. We have updated our manuscript to address the concerns and suggestions you brought forth in your reviews and include a summary of all the changes here.

### Summary
Our updated manuscript contains the following updates from our original submission:
1. We have updated several of our definitions, as well significant parts of our text, to make sure that our ideas are communicated in a more succinct and clear manner.
1. As per reviewers 7LHy and yfkD request, we have updated our paper to include three new baselines: VAE, $\beta$-VAE, and Self-supervised Neural Neworks (SENN). We are delighted to say that this has led to our metrics highlighting new surprising insights and we discuss such insights with more detail in our experimentation and discussion sections. In summary, we found that unsupervised VAEs were able to extract purer, and higher-quality, representations that those learnt with weak-supervision. Similarly, we found that SENN, as was the case for CCD, can potentially learn concept representations whose quality matches that of representations learnt with full concept supervision (e.g., Concept Whitening).
1. We have slightly updated our abstract to include these new findings in its conclusion.
1. We have updated our experiments and discussion sections to make the distinction between expected and novel results clearer.
1. We have updated our OIS definitions to justify why we make use of AUC as a proxy to mutual information and why we opted to use the Frobenius norm of the difference between the oracle and purity matrices as a similarity metric.
1. We provide a new empirical justification for our niching metrics based on a real-world task (Appendix 6.6).
1. We include new experiments showing the effect that capacity has in the encoder and decoder models independently of each other (Appendix 6.7).
1. We include further details on how to select a value of $\beta$ for evaluating niching metrics in Appendix 6.5.
1. We have added a new Appendix (6.3) clarifying the details of how purity matrices are computed.
1. We have added a new Appendix (6.4) clarifying the details of how concept alignments are computed when evaluating OIS if a method was not provided with explicit concept supervision.

---

### Decision · Program_Chairs · 2022-01-20

**Decision:**

Reject

**Comment:**

This paper considers the question of whether recent concept-based learning algorithms, as well disentangled representation learning algorithms, result in high-quality representations. In particular, the authors consider what high-quality should mean in terms of the relationship with ground truth concepts and the ability to make accurate predictions for a downstream task. To this end, they propose two main metrics for representations that are explicitly or implicitly encouraged to encode concepts. While the premise of this paper has been appreciated by the reviewers, some concerns about the details of the metrics proposed and experimental results which have been raised by the reviewers remain post rebuttal. Given this, we are unable to recommend the acceptance of the paper at this time. We hope the authors find the reviewer feedback useful.